# CONTINUAL PRE-TRAINING OF LANGUAGE MODELS

**Zixuan Ke**[1*], **Yijia Shao**[2*], **Haowei Lin**[2*], **Tatsuya Konishi**[3†], **Gyuhak Kim**[1], **and Bing Liu**[1‡]

[1]Department of Computer Science, University of Illinois at Chicago
[2]Wangxuan Institute of Computer Technology, Peking University
[3]KDDI Research
[1]{zke4,gkim87,liub}@uic.edu
[2]{shaoyj,linhaowei}@pku.edu.cn
[3]tt-konishi@kddi-research.jp

## ABSTRACT

Language models (LMs) have been instrumental for the rapid advance of natural language processing. This paper studies continual pre-training of LMs, in particular, *continual domain-adaptive pre-training* (or *continual DAP-training*). Existing research has shown that further pre-training an LM using a domain corpus to adapt the LM to the domain can improve the end-task performance in the domain. This paper proposes a novel method to *continually DAP-train* an LM with a sequence of unlabeled domain corpora to adapt the LM to these domains to improve their end-task performances. The key novelty of our method is a soft-masking mechanism that directly controls the update to the LM. A novel proxy is also proposed to preserve the general knowledge in the original LM. Additionally, it contrasts the representations of the previously learned domain knowledge (including the general knowledge in the pre-trained LM) and the knowledge from the current full network to achieve knowledge integration. The method not only overcomes *catastrophic forgetting*, but also achieves *knowledge transfer* to improve end-task performances. Empirical evaluation demonstrates the effectiveness of the proposed method.[1]

## 1 INTRODUCTION

Pre-trained language models (LMs) like BERT (Devlin et al., 2019) and RoBERTa (Liu et al., 2019) have significantly advanced NLP. Recently, LMs have also been used by many *continual learning* (CL) systems to learn a sequence of end-tasks incrementally (Ke et al., 2021a; Sun et al., 2020; Huang et al., 2021), which we call *continual end-task learning*. It is also desirable to continually pre-train LMs themselves. This includes (1) *continual general pre-training*, which incrementally updates the LM using the most recent data that has a similar distribution as the pre-training data, and (2) *continual domain-adaptive pre-training*, which further pre-trains a LM incrementally to adapt it to a sequence of domains. Note that *LM editing* (with or without continual learning) (Mitchell et al., 2022) that corrects mistakes learned in the LM is a special case of continual end-task learning (Kim et al., 2022) as each editing task or a group of editing tasks learned together is basically a *task* in continual learning, which aims to perform the editings correctly without interfering with or forgetting the other knowledge already learned in the current LM.

This paper focuses on *continual **d**omain-**a**daptive **p**re-training* (or *continual DAP-training*) of LMs. It is known that *DAP-training*[2] an LM (without continual learning) using a large unlabeled domain corpus before end-task fine-tuning achieves better results (Gururangan et al., 2020; Xu et al., 2019; Ke et al., 2022b). This paper goes a step further to continually learn to improve an LM's ability to handle new or emerging domains or topics *without* forgetting the skills or knowledge learned in the past. This is important in the real world, where the data shifts constantly and new domains, events or topics keep emerging (Ke et al., 2022b) and the LM needs to be updated to serve the users better.

---

[*]Equal contribution
[†]The work was done when this author was visiting Bing Liu's group at University of Illinois at Chicago.
[‡]Correspondance author. Bing Liu <liub@uic.edu>
[1]The code is available at https://github.com/UIC-Liu-Lab/ContinualLM
[2]Depending on different contexts or authors, DAP-training is also called *post-training* or *pre-finetuning*.

We call this problem *continual DAP-training*. Starting from a pre-trained general LM (i.e., the LM has already been pre-trained on $D_0$), we incrementally DAP-train a sequence of domain corpora $D_1, D_2, ....$ Once a domain is trained, its data is no longer accessible. This is different from conventional *continual learning* (**CL**) where each task is an end-task. In the proposed continual DAP-training, each task is an unlabeled domain corpus to be learned. An end-task fine-tunes the continually DAP-trained LM to evaluate its performance. It is worth noting that $D_0$ is usually a *broad or general domain* (e.g., News). In practice, a continually DAP-trained LM may be trained by individual users, institutions or a mix of both who have one or more large corpora of some particular domains. In such cases, the raw data may not be shared, but the final LM can be shared by all.

There are multiple desiderata for a continual DAP-training system: **(1)** It should not suffer from *catastrophic forgetting* (**CF**), i.e., it should perform reasonably well on learned domains. This requires the system (a) to overcome CF for each new domain and (b) to overcome CF for the general language knowledge in the LM. This is important because the knowledge learned from each domain alone will not be sufficient for good end-task performances. **(2)** It should encourage knowledge transfer (**KT**) across domains to achieve improved end-task performances. This requires the system to enable (a) *forward transfer*, learning a new domain by leveraging the knowledge from previous domains, and (b) *backwards transfer*, gaining improved performance on previous domains after learning a relevant new domain. **(3)** It should work without requiring the domain-ID for each end-task fine-tuning.

None of the existing CL methods can achieve all the above. This paper represents a step towards achieving them. The proposed method is called **DAS** (Continual *DA*-pre-training of LMs with *S*oft-masking). DAS proposes a novel *soft-masking mechanism* that computes the importance (a real number between 0 and 1) of units[3] for general or domain knowledge and soft-mask them based on their importance values to control the backward gradient flow. In the forward pass, soft-masking is not applied, which encourages KT across domains. It does not isolate any sub-network for any domain so that the knowledge in the full LM can be leveraged for end-task fine-tuning.

To apply this mechanism, DAS implements two functions: (1) *Initialization*, which computes the importance of units to **the general knowledge** in the LM without accessing the LM pre-training data ($D_0$). It is applied on the pre-trained LM before the continual learning starts, and (2) *continual learning*, which DAP-trains each domain while preventing CF on the general and domain knowledge and encouraging cross-domain KT. In (1), it is not obvious how to compute the importance without pre-training data. DAS proposes a novel proxy based on *robustness* to compute the importance of units for the general knowledge. In (2), the soft-masking is directly applicable because we have the domain data and the importance can be computed based on its gradient inspired by the pruning community (Li et al., 2021; Michel et al., 2019). Moreover, DAS contrasts the previously learned knowledge and the full (including both the learned domains and the current domain) knowledge to encourage the current domain representation to learn knowledge that is not already in the knowledge learned from previous domains and integrate it with the learned knowledge[4]. In end-task fine-tuning, DAS does not requires the domain-ID as all knowledge is accumulated into the DAP-trained LM.

In summary, this work makes the following contributions. **(i)** It studies the new problem of *continual DAP-training* and discovers that the full LM is needed for a good continual DAP-training method. The popular *parameter-isolation approach* to overcoming CF in convention CL is unsuitable. **(ii)** It proposes a novel soft-masking method to overcome CF and to encourage KT, and a constrative learning based method for knowledge integration. **(iii)** To preserve the general knowledge in the LM, a novel proxy is also proposed. **(iv)** Experimental results demonstrate the effectiveness of DAS.

## 2 RELATED WORK

**DAP-training.** DAP-training can be achieved by directly updating the LM (Xu et al., 2019; Sun et al., 2019; Lee et al., 2020; Alsentzer et al., 2019; Gururangan et al., 2020; Chakrabarty et al., 2019; Ke et al., 2022b) or by training only a small set of additional parameters. For example, Pfeiffer et al. (2020); Wang et al. (2020a); Ke et al. (2021a;b;c) trained adapters and Gu et al. (2021) trained a prompt to adapt to a domain. While adapter and prompt could be effective, transfer knowledge among

---

[3]For simplicity, we use the term *units* to mean both *attention heads* and *neurons*.

[4]Contrasting the past domains and only the domain-specific knowledge gives poorer results (see Sec. 4.2) as it causes the two types of knowledge to split rather than to integrate.

these additional modules is usually challenging and can be inaccurate. DAS belongs to the former family that directly updates the LM. This is very challenging for CL due to CF. To our knowledge, no existing system in this family is about CL.

**Continual learning.** Most CL methods were proposed to overcome CF: (1) *Regularization methods* (Kirkpatrick et al., 2016; Seff et al., 2017) compute the importance of each parameter to previous tasks and uses a regularizer to penalize the sum of changes. DAS is related to but also very different from EWC (Kirkpatrick et al., 2016). (1) DAS does not control each parameter/weight, but only attention heads or neurons based on their importance scores. This gives less forgetting (see the forgetting rate in Table 2) because even a small change to each parameter for a neuron can give a large total change to the neuron's activation. (2) DAS directly controls the backward gradient flow on each neuron, which is more fine-grained and effective than the sum of changes of all parameters. Our experimental results confirm that EWC is significantly poorer than DAS (see Table 2). (2) *Replay methods* retain (Rebuffi et al., 2017; Wang et al., 2020b) or generate some data of old tasks (Shin et al., 2017; He & Jaeger, 2018) and use them in learning a new task; (3) *parameter-isolation methods* (Serrà et al., 2018; Wortsman et al., 2020) allocate neurons and parameters or sub-networks for different tasks/domains and mask them in task learning. For continual DAP-training, this means that end-tasks cannot use the general knowledge in the LM, which results in poor end-task performances.

In NLP, CL has been used for slot filling (Shen et al., 2019), language learning (Li et al., 2019), sentiment analysis (Ke et al., 2021a), topic modeling (Gupta et al., 2020), question answering (Greco et al., 2019) and text classification (Sun et al., 2020; Huang et al., 2021; Chuang et al., 2020). But none is for DAP-training. Some recent CL papers concern LMs. The system in (Madotto et al., 2020) learns separate adapters for different domains and thus has no CF or KT. DEMIX (Gururangan et al., 2021) initializes the new adapter with the closest old adapter. CPT (Ke et al., 2022a) and ELLE (Qin et al., 2022) are most closely related to DAS. However, CPT uses the parameter-isolation approach to learn and protect each task, which is weak (see Sec. 4.2). It also needs domain-ID in end-task fine-tuning. ELLE has to start from pre-training the LM itself rather than from a pre-trained LM like DAS. It also uses a large memory (1G per domain) to store the replay data (including the pre-training data) and expands the network for each domain. Neither is required in DAS. Jin et al. (2021) evaluated several existing CL techniques in a similar setting as DAS and performed analyses on dealing with CF. However, no new technique was proposed in the paper.

**Neural network pruning.** Many parameters in a network are redundant and can be pruned (Li et al., 2021; Lai et al., 2021; Michel et al., 2019; Voita et al., 2019). Existing methods include discarding parameters with small absolute values (Han et al., 2015; Guo et al., 2016), accumulated gradient (Michel et al., 2019), and lottery ticket hypothesis (Brix et al., 2020). However, these methods are not directly applicable as we need to preserve not only individual domain knowledge but also the general knowledge in the LM. For general knowledge, since we do not have any pre-training data, a proxy is proposed based on robustness. For domain knowledge, we adopt a pruning method but use the importance as soft-masks as we want to accumulate knowledge rather than to compress the LM.

**Contrastive Learning.** Contrastive learning (Chen et al., 2020; He et al., 2020) learns good representations by maximizing the similarity of positive pairs and minimizes that of negative pairs,

$$\mathcal{L}_{\text{contrast}} = -\frac{1}{N} \sum_{n=1}^{N} \log \frac{e^{(\text{sim}(q_n, q_n^+)/\tau)}}{\sum_{j=1}^{N} e^{(\text{sim}(q_n, q_j^+)/\tau)}}, \tag{1}$$

where $N$ is the batch size, $\tau$ is a temperature parameter, $\text{sim}(\cdot)$ is a similarity metric, and $q_n$ and $q_n^+$ are representations for positive pairs $x_n$ and $x_n^+$. DAS contrasts the learned knowledge from previous domains and the pre-trained LM (general knowledge) with the full knowledge (including both the previous domains and current domain knowledge) to achieve a complementary effect.

## 3 PROPOSED DAS TECHNIQUE

Continual DAP-training in DAS is based on two main ideas: (1) preserving the important general language knowledge in the LM and the knowledge learned from previous domains to overcome CF by soft-masking units based on their importance, which also facilitates cross-task knowledge transfer (KT), and (2) encouraging the model to learn complementary representations of the current domain and previous domains to achieve knowledge integration. Figure 1 gives an overview of DAS.

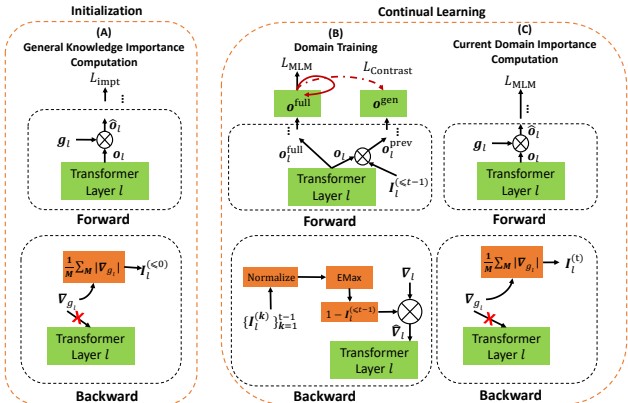

Figure 1: Illustration of DAS. The red cross indicates that the gradient is not used to update the Transformer but only to compute importance. (A) **Initialization** (Sec. 3.1) computes the importance of units for the general knowledge in the LM. (B) **Domain Training** (Sec. 3.2) trains a new domain using the importance scores as soft-masks and contrasts the previously learned knowledge and the full knowledge. (C) **Importance Computation** (Sec. 3.3) computes the importance of the units for the current domain.

The whole learning consists of two main functions: (i) *initialization* and (ii) *continual learning*. (i) computes the importance of units to the general language knowledge in the LM. It is done before the continual learning starts. (ii) is for continual learning, which consists of two steps: (a) *domain training* and (b) *importance computation*. (a) takes the importance scores accumulated so far (including those to the general knowledge in the original LM and to the knowledge learned from previous domains) and the input data of the current domain to learn the domain and to achieve (1) and (2) above, while (b) computes the importance scores for the current domain for future use. The following sub-sections present each function and step in detail.

## 3.1 INITIALIZATION: COMPUTING IMPORTANCE OF UNITS TO THE GENERAL KNOWLEDGE

This initialization function computes the importance of units (attention heads and neurons) in the Transformer for the *general* knowledge in the original LM. The key components of a Transformer are *multi-head attention layer*, *intermediate layer* and *output layer*. Below, we use "layer" or $l$ to indicate any of these three layers because our method treats the three layers similarly.

**Importance of units in a layer.** It has been found that not all units in a layer are important (Michel et al., 2019). We introduce a *virtual parameter*, $g_l$, for computing the importance of the units in a layer $l$. We call these *virtual* parameters as each $g^{(k)}$ is initialized to 1. We only need the gradient on each parameter to compute the importance of its corresponding unit, no update to any parameter.

$$\hat{o}_l = g_l \otimes o_l, \tag{2}$$

where $o_l$ refers to the output of layer $l$ (which can be any of the three layers mentioned above). The $\otimes$ refers to element-wise multiplication, i.e., each variable $g_{l,i}$ in $g_l$ corresponding to a unit (a neuron or attention head) in the layer. We adapt the gradient-based importance detection method in (Michel et al., 2019) for our purpose. Given a dataset $D = \{(x_n, y_n)\}_{n=1}^{N}$ of $N$ samples ($y_n$ is the class label of $x_n$ as (Michel et al., 2019) worked on supervised learning), the importance of neurons or heads in the layer is estimated with a gradient-based proxy score

$$I_l = \frac{1}{N} \sum_{n=1}^{N} |\frac{\partial \mathcal{L}_{\text{impt}}(x_n, y_n))}{\partial g_l}|, \tag{3}$$

where $\mathcal{L}_{\text{impt}}$ is a task-specific loss function. Note the virtual parameter $g_l$ is initialized as all 1's, and is not changed. This is because we need only its average gradient $\nabla_{g_l}$ (the term within $||$ in Eq. 3) over all the data to compute the importance and will not use the gradient to update the virtual parameter. In training (Sec. 3.2 and Fig 1 (B)), the virtual parameter can be discarded. The resulting $\bar{I}_l$ is of the same size as $g_l$, each entry corresponding to the importance of a unit (a neurons or attention head).

Recall that the *initialization* function is to learn the importance of units to the *general knowledge* in the LM (denoted as $\boldsymbol{I}_l^{(0)}$). Although Eq. 3 offers a possible way, it is not directly applicable. If we use the domain data at hand and employ the MLM loss as $\mathcal{L}_{\text{impt}}$, $\nabla_{\boldsymbol{g}_l}$ only gives the importance for the *domain-specific* knowledge. However, to compute the importance of units to the general knowledge in the LM (which is our goal), we need the original data used in pre-training the LM to compute the $\mathcal{L}_{\text{impt}}$. In practice, such data is not accessible to users of the LM. Further, label is needed in Eq. 3 but our domain corpus is unlabeled in DAP-training. To address these issues, we propose a *proxy KL-divergence loss* ($\mathcal{L}_{\text{proxy}}$) to replace $\mathcal{L}_{\text{impt}}$ to learn units importance for the general knowledge.

**Proxy KL-divergence loss.** We propose to use model *robustness* as the proxy, i.e., we try to detect units that are important for LM's *robustness*. Their gradients, $\nabla_{\boldsymbol{g}_l}$, then indicate the robustness and the importance to the LM model. Our rationale is as follows: If an $I_{l,i}^{(0)}$ (the importance of unit $i$ in layer $l$) has a high value, then it is important to the LM's robustness because its change can cause the LM to change a lot. It is thus an important unit. In contrast, if $I_{l,i}^{(0)}$ is small, it is a less important unit. To compute the *robustness* of the LM, we take a subset of the current domain data $\{\boldsymbol{x}_n^{\text{sub}}\}$[5] (no label in DAP-training) and input $\boldsymbol{x}_n^{\text{sub}}$ twice to the LM to obtain two representations of it and then compute the KL-divergence between them,

$$\mathcal{L}_{\text{impt}} = \text{KL}(f_{\text{LM}}^1(\boldsymbol{x}_n^{\text{sub}}), f_{\text{LM}}^2(\boldsymbol{x}_n^{\text{sub}})), \tag{4}$$

where $f_{\text{LM}}^1$ and $f_{\text{LM}}^2$ are the LM with different dropout masks. We don't need to add any additional dropouts to implement these two as the Transformer already has dropout masks placed on fully-connected layers and attention probabilities. Thus, simply feeding the same input to the Transformer twice will get two representations with different dropout masks. Since dropout is similar to adding noise, the difference between the two representations can be regarded as the *robustness* of the LM.

## 3.2 Training: Learning a New Domain via Soft-masking and Contrastive Loss

Recall we want to preserve the learned knowledge in the LM during DAP-training using the accumulated importance $\boldsymbol{I}_l^{(\leq t-1)}$ when we learn domain $t$, which includes both the importance for the general knowledge $\boldsymbol{I}_l^{(0)}$ (Sec. 3.1) and learned domain-specific knowledge $\boldsymbol{I}_l^{(k)}$ of each domain $k$ ($k$ can be any domain in $\{1...t-1\}$) that has been learned (Sec. 3.3). This is achieved by soft-masking the learning based on accumulated importance as follows.[6]

**Accumulating importance.** We accumulate the importance after task $t-1$ was learned is done via element-wise max (EMax) as follows:

$$\boldsymbol{I}_l^{(\leq t-1)} = \text{EMax}(\{\boldsymbol{I}_l^{(t-1)}, \boldsymbol{I}_l^{(\leq t-2)}\}), \tag{5}$$

where $t$ refers to the current task-ID and $\boldsymbol{I}_l^{(\leq t-2)}$ refers to the previously accumulated importance at task $t-2$. We do not need to save $\boldsymbol{I}_l^0$ and all $\{\boldsymbol{I}_l^{(k)}\}_{k=1}^{t-1}$ for Eq. 5. We only save the incrementally accumulated importance after training of each task.

**Soft-masking units.** Given the accumulated importance $\boldsymbol{I}_l^{(\leq t-1)}$ of layer $l$ and the DAP-training loss $\mathcal{L}_{\text{DAP-train}}$ (typically the MLM loss; we also propose an additional loss in Eq. 7), we constrain (or soft-mask) its corresponding gradient ($\nabla_l$) flow as follows,

$$\hat{\nabla}_l = (1 - \boldsymbol{I}_l^{(\leq t-1)}) \otimes \nabla_l, \tag{6}$$

As mentioned in Sec. 3.1, we expand (by copying) the importance $\boldsymbol{I}_l^{(\leq t-1)}$ to match the dimensions of $\nabla_l$ to apply it to all associated parameters. This is *soft-masking* as each element in $\boldsymbol{I}_l^{(\leq t-1)}$ is a real number in $[0, 1]$ (not binary $\{0, 1\}$), which gives the model the flexibility to adjust any unit.

---

[5] We use a subset to save computation as we assume that the DAP-training domain can be very large. In Sec. 4, we show that a subset is sufficient to compute the importance of units for the general knowledge.

[6] Before training, we normalized the the importance values in each layer $l$ for a domain $k$ by making the importance scores for all units in the layer having a mean of 0 and standard deviation of 1. To further facilitate soft-masking, the normalized importance scores are rounded by a `Tanh` activation so that the values are in the interval of [0,1]. To simplify the notation, we still use $\boldsymbol{I}_l^{(k)}$ to represent the resulting importance.

| Unlabelde Domain Datasets | | | End-Task Classification Datasets | | | | |
|---|---|---|---|---|---|---|---|
| Source | Dataset | Size | Dataset | Task | #Training | #Testing | #Classes |
| Reviews | Yelp Restaurant | 758MB | Restaurant | Aspect Sentiment Classification (ASC) | 3,452 | 1,120 | 3 |
| | Amazon Phone | 724MB | Phone | Aspect Sentiment Classification (ASC) | 239 | 553 | 2 |
| | Amazon Camera | 319MB | Camera | Aspect Sentiment Classification (ASC) | 230 | 626 | 2 |
| Academic Papers | ACL Papers | 867MB | ACL | Citation Intent Classification | 1,520 | 421 | 6 |
| | AI Papers | 507MB | AI | Relation Classification | 2,260 | 2,388 | 7 |
| | PubMed Papers | 989MB | PubMed | Chemical-protein Interaction Prediction | 2,667 | 7,398 | 13 |

Table 1: Statistics of datasets for DAP-training. More details of their end-task supervised learning datasets are given in Appendix A.

We note that the above soft-masks are only applied in the backward pass, but not in the forward pass, which encourages knowledge transfer as each domain training can leverage the knowledge learned from all past domains. To further encourage the model to learn a good representation from both the accumulated knowledge ($I_l^{(\leq t-1)}$) and the full knowledge (both accumulated and current domain knowledge), we introduce a contrastive learning method to encourage complementary representation.

**Integrating the previously learned knowledge and the current domain knowledge.** Soft-masking helps prevent forgetting the previously learned knowledge. We want to further encourage knowledge transfer by integrating the new and learned knowledge. We propose to contrast the previously learned knowledge and the full knowledge (both previously learned knowledge and the current domain knowledge). Note that the contrasting cannot do anything to the shared past knowledge as it is protected by soft-masks. Thus, it effectively pushes the current domain knowledge away to be complementary to the past knowledge. This is done based on the current domain data as follows.

***Contrasting the learned and full knowledge.*** We denote the output of LM without any consideration of importance as $o^{\text{full}}$, which refers to the full knowledge. We further denote the output of LM that is multiplied by the importance (i.e., $I_l^{(\leq t-1)} \otimes o_l$) as $o^{\text{prev}}$, which refers to the previously learned knowledge. We contrast the two by using $o^{\text{full}}$ as anchor and $o^{\text{full}}$ with different dropouts as positive samples (dentoed as $o^{\text{full+}}$). $o^{\text{prev}}$ is used as negative instances.

Formally, with $o_n^{\text{full}}$, $o_n^{\text{full+}}$, and $o_n^{\text{prev}}$, our contrastive loss is ($\text{sim}(\cdot)$ is the cosine similarity),

$$\mathcal{L}_{\text{contrast}} = -\frac{1}{N} \sum_{n=1}^{N} \log \frac{e^{\text{sim}(o_n^{\text{full}}, o_n^{\text{full+}})/\tau}}{\sum_{j=1}^{N} (e^{\text{sim}(o_n^{\text{full}}, o_j^{\text{full+}})/\tau} + e^{\text{sim}(o_n^{\text{full}}, o_j^{\text{prev}})/\tau})}. \tag{7}$$

Compared to Eq. 1, the second term is added in the denominator, i.e., representations in the previously learned knowledge as additional negative instances. Figure 1 (B) shows a red arrow pointed from $o^{\text{full}}$ to itself, indicating the positive instances are from inputting twice. The dashed red arrow pointing to $o^{\text{prev}}$ indicates the negative instances contrasting the full and previously learned knowledge.

**Final Loss Function**. The final DAP-training loss combines the Masked Language Model (MLM) loss after applying the proposed soft-masking for the general knowledge (Sec. 3.1) and the proposed contrastive loss ($\lambda$ is a hyper-parameter),

$$\mathcal{L}_{\text{DAP-train}} = \mathcal{L}_{\text{MLM}} + \lambda \mathcal{L}_{\text{contrast}} \tag{8}$$

### 3.3 COMPUTE IMPORTANCE OF UNITS TO THE CURRENT DOMAIN

After training the new/current domain $t$, we learn the units importance by applying Eq. 3 for the domain. We do not need any proxy to compute $\mathcal{L}_{\text{impt}}$ as in Eq. 4 because we can directly use the current domain data. Specifically, we randomly sample a subset (a hyper-parameter) of the current domain data $\{(x_n^{\text{sub}}, y_n^{\text{sub}})\}$, where $x_n^{\text{sub}}$ is the input and $y_n^{\text{sub}}$ is the masked token as in MLM self-supervised loss. We can then easily compute the importance $I_l^{(t)}$ by plugging $\mathcal{L}_{\text{MLM}}$ into $\mathcal{L}_{\text{impt}}$ in Eq. 3. The resulting $I_l^{(t)}$ will be used in the next task by accumulating with the previously accumulated importance (Eq. 5) and soft-masking the learning (Eq. 6).

## 4 EXPERIMENTS

We use RoBERTa (Liu et al., 2019)[7] as the LM. Following the standard evaluation setup (Lange et al., 2019) and, after a domain is trained, its training data is discarded. After all domains are incrementally learned, the final model is evaluated by fine-tuning the end-tasks in all domains.

---

[7] https://huggingface.co/roberta-base

## 4.1 Datasets and Baselines

**Datasets:** Table 1 shows the statistics of the 6 *unlabeled domain corpora* for DAP-training and their 6 corresponding *end-task classification datasets*.[8] 3 of them are about reviews: *Yelp Restaurant* (Xu et al., 2019), *Amazon Phone* (Ni et al., 2019), *Amazon Camera* (Ni et al., 2019); 3 of them are academic papers: *ACL Papers* (Lo et al., 2020), *AI Papers* (Lo et al., 2020), and *PubMed Papers*[9]. Their corresponding *end-task classification datasets* are:[10] *Restaurant*[11], *Phone* (Ding et al., 2008; Hu & Liu, 2004), *Camera* (Ding et al., 2008; Hu & Liu, 2004), *ACL* (ACL-ARC in (Jurgens et al., 2018)), *AI* (SCIERC in (Luan et al., 2018)), and PubMed (CHEMPORT in (Kringelum et al., 2016)).

**Baselines.** We use 16 baselines, including non-continual learning (Non-CL) and continual learning (CL) baselines. All CL baselines are originally for learning supervised data except DEMIX. We adapt them and replace their backbone with RoBERTa. Details of each baseline is given in Appendix B.

**Non-CL Baselines**: Each baseline here builds a separate model for each task. **(1) Pool.** We pool the data of all domains together and train only one model for all domains. **(2) RoBERTa** (Liu et al., 2019) uses RoBERTa for end-task fine-tuning without DAP-training. **(3) DAP-RoBERTa** uses the existing DAP-training method (MLM) in (Gururangan et al., 2020) to posst-train each domain separately. **(4) DAP-Adapter** adds adapter layers in Transformer for each domain for DAP-training (Jang et al., 2021; Madotto et al., 2020; Houlsby et al., 2019). Only the added adapters are trainable. In end-task fine-tuning, both RoBERTa and the adapters are trainable. **(5) DAP-Prompt** is from (Lester et al., 2021). In DAP-training, RoBERTa (the LM) is fixed and only the prompts are trained. In end-task fine-tuning, both the LM and the trained prompt are trainable.

**CL Baselines**: We use 2 naive baselines, which keep learning more domains with no mechanism to deal with CF or transfer. **(6) NCL** (Naive CL) continually DAP-trains the RoBERTa; and **(7) NCL-Adapter** continually DAP-trains a set of adapters (Houlsby et al., 2019).

8 baselines are CL systems: **(8) DEMIX** (Gururangan et al., 2021) adds a new adapter for each new domain and initializes it with a previous adapter nearest to the new domain; **(9) BCL** (Ke et al., 2021c) uses capsule networks. **(10) CLASSIC** (Ke et al., 2021b) uses contrastive learning. **(11) KD** is knowledge distillation (Hinton et al., 2015). **(12) EWC** (Buzzega et al., 2020) is a popular regularization-based method. **(13) DER++** (Buzzega et al., 2020) is a replay method based on knowledge distillation. 16.4K tokens are saved for each domain in the replay memory, which is the largest memory we can use for the system to run. **(14) HAT** (Serrà et al., 2018) is an effective *parameter-isolation* method. HAT is applied to Transformer layers (i.e., self-attention, intermediate and output layers). **(15) HAT-All** is a HAT variant that uses all features *from the LM* to do end-tasks (instead of only features from its domain sub-network as in HAT). **(16) HAT-Adapter** (Ke et al., 2021c) uses HAT within adapters. ELLE (Qin et al., 2022) is not included as we adapted it for our purpose by learning from RoBERTa, but it fails to converge.

## 4.2 Results Analysis and Ablation Study

Due to space limits, ***Implementation Details*** are given in Appendix C. Table 2 reports the end-task fine-tuning results of all 15 systems on the 6 datasets. We can see that the proposed DAS outperforms all baselines on average and also achieve the best knowledge transfer (negative forgetting rate).

(1) DAS is slightly better than Pool on average. This may be because (a) some domains are quite different (e.g. camera reviews and ACL papers), which results in some negative transfer in Pool. (b) DAS can learn with the general and previous domain knowledge protected by soft-masks.

(2). DAS achieves both forgetting prevention and knowledge transfer. Those baselines (KD, EWC, DER++) focusing only on forgetting prevention give poorer performance as they sacrifice accuracy to avoid CF. Those baselines (BCL, CLASSIC and DEMIX) perform knowledge transfer achieve better results but still poorer than DAS. DEMIX has very weak transfer. BCL, which can avoid CF while

---

[8]We down-sampled the *PubMed* due to its huge original size. In general, our datasets are much smaller than those used in (Gururangan et al., 2020) (which used more than 11GB of data for each domain). Our experiments showed that a smaller dataset is sufficient and more data does not help. It also requires less computing power.

[9]https://pubmed.ncbi.nlm.nih.gov/

[10]Our results are different from those presented in Table 5 of (Gururangan et al., 2020) because we observe very high variances due to very small test sets and thus enlarge the test set and reduce the training set slightly.

[11]https://alt.qcri.org/semeval2014/task4/

| Category | Domain | Restaurant | | ACL | | AI | | Phone | | PubMed | Camera | | Average | | Forget R. | |
|---|---|---|---|---|---|---|---|---|---|---|---|---|---|---|---|---|
| | Model | MF1 | Acc | MF1 | Acc | MF1 | Acc | MF1 | Acc | MF1 | MF1 | Acc | MF1 | Acc | MF1 | Acc |
| | Pool | 80.96 | 87.80 | 69.69 | 74.11 | 68.55 | 75.97 | 84.96 | 86.95 | 73.34 | 86.03 | 90.83 | 77.25 | 81.50 | — | |
| Non-CL | RoBERTa | 79.81 | 87.00 | 66.11 | 71.26 | 60.98 | 71.85 | 83.75 | 86.08 | 72.38 | 78.82 | 87.03 | 73.64 | 79.27 | — | |
| | DAP-RoBERTa | 80.84 | **87.68** | 68.75 | 73.44 | 68.97 | 75.95 | 82.59 | 85.50 | 72.84 | 84.39 | 89.90 | 76.40 | 80.89 | — | |
| | DAP-Adapter | 80.19 | 87.14 | 68.87 | 72.92 | 60.55 | 71.38 | 82.71 | 85.35 | 71.68 | 83.62 | 89.23 | 74.60 | 79.62 | — | |
| | DAP-Prompt | 79.00 | 86.45 | 66.66 | 71.35 | 61.47 | 72.36 | 84.17 | 86.53 | **73.09** | 85.52 | 90.38 | 74.98 | 80.03 | — | |
| | NCL | 79.52 | 86.54 | 68.39 | 72.87 | 67.94 | 75.71 | 84.10 | 86.33 | 72.49 | 85.71 | 90.70 | 76.36 | 80.77 | 1.14 | 1.05 |
| | NCL-Adapter | 80.13 | 87.05 | 67.39 | 72.30 | 57.71 | 69.87 | 83.32 | 85.86 | 72.07 | 83.70 | 89.71 | 74.05 | 79.48 | 0.15 | -0.02 |
| | DEMIX | 79.99 | 87.12 | 68.46 | 72.73 | 63.35 | 72.86 | 78.07 | 82.42 | 71.73 | 86.59 | 91.12 | 74.70 | 79.66 | 0.74 | 0.36 |
| | BCL | 78.97 | 86.52 | **70.71** | **74.58** | 66.26 | 74.55 | 81.70 | 84.63 | 71.99 | 85.06 | 90.51 | 75.78 | 80.46 | -0.06 | -0.19 |
| | CLASSIC | 79.89 | 87.05 | 67.30 | 72.11 | 59.84 | 71.08 | 84.02 | 86.22 | 69.83 | 86.93 | 91.25 | 74.63 | 79.59 | 0.44 | 0.25 |
| CL | KD | 78.05 | 85.59 | 69.17 | 73.73 | 67.49 | 75.09 | 82.12 | 84.99 | 72.28 | 81.91 | 88.69 | 75.17 | 80.06 | -0.07 | 0.01 |
| DAP-train | EWC | **80.98** | 87.64 | 65.94 | 71.17 | 65.04 | 73.58 | 82.32 | 85.13 | 71.43 | 83.35 | 89.14 | 74.84 | 79.68 | 0.02 | -0.01 |
| | DER++ | 79.00 | 86.46 | 67.20 | 72.16 | 63.96 | 73.54 | 83.22 | 85.61 | 72.58 | 87.10 | 91.47 | 75.51 | 80.30 | 2.36 | 1.53 |
| | HAT | 76.42 | 85.16 | 60.70 | 68.79 | 47.37 | 65.69 | 72.33 | 79.13 | 69.97 | 74.04 | 85.14 | 66.80 | 75.65 | -0.13 | -0.29 |
| | HAT-All | 74.94 | 83.93 | 52.08 | 63.94 | 34.16 | 56.07 | 64.71 | 74.43 | 68.14 | 65.54 | 81.44 | 59.93 | 71.33 | 3.23 | 1.83 |
| | HAT-Adapter | 79.29 | 86.70 | 68.25 | 72.87 | 64.84 | 73.67 | 81.44 | 84.56 | 71.61 | 82.37 | 89.27 | 74.63 | 79.78 | -0.23 | -0.18 |
| | DAS | 80.34 | 87.16 | 69.36 | 74.01 | **70.93** | **77.46** | **85.99** | **87.70** | 72.80 | **88.16** | **92.30** | **77.93** | **81.91** | **-1.09** | **-0.60** |

Table 2: End-task macro-F1 (MF1), accuracy and forgetting rate results for all domains *after the continual DAP-training of all domains*, except for CHEMPORT in the PubMed domain, for which we use micro-F1 following (Gururangan et al., 2020; Dery et al., 2021; Beltagy et al., 2019). The results are averages of 5 random seeds (the domain training order is as they appear in the first row). Due to space limits, the results for *different domain orders* and the *standard deviations* are reported in Appendix D and Appendix E, respectively). Non-CL baselines have no forgetting.

also achieving some transfer, is weaker than NCL. In general, CL baselines are all poorer than DAS as they don't have methods to encourage knowledge transfer or they have to rely on adapters.

(3). Directly learning the domains within the LM helps DAS achieve better results than adapter and prompt based methods. DAS is better than adapter-based systems (DAP-Adapter, NCL-Adapter and HAT-Adapter) and prompt-based system (DAP-Prompt). This is because adapters and prompts do not have sufficient trainable parameters, which are also randomly initialized and can be hard to train.

(4). Using the full LM to learn all tasks rather than using sub-networks (of HAT-based methods) makes DAS more effective. HAT performs poorly, indicating it is unsuitable for DAP-training as discussed in Sec. 1. Even if we use all features (not only the feature from the corresponding sub-network), we still get poor results (HAT-All) as the features used in DAP-training (in an LM sub-network) are different from features used in end-task fine-tuning (features from the whole LM).

**Knowledge transfer and forgetting avoidance.** To see how the models fare on CF and knowledge transfer, we compare the forgetting rates (**forget R.**) (Liu et al., 2020), $\frac{1}{t-1}\sum_{k=1}^{t-1} A_{k,k} - A_{t,k}$, where $A_{k,k}$ is the end-task accuracy right after its domain $k$ is DAP-trained, and $A_{t,k}$ is the accuracy of the end-task of domain $k$ after DAP-training the last domain $t$. We average over all end-tasks except the last as the last domain has no forgetting. The higher the forgetting rate is, the more forgetting it has. Negative rates indicate positive knowledge transfer. Clearly, DAS has the strongest negative forgetting rate, indicating it does well on both forgetting prevention and knowledge transfer. NCL, NCL-Adapter, DEMIX, EWC, KD and DER++ all suffer from some forgetting. HAT has no forgetting but it cannot learn well. HAT and BCL have no forgetting but are weak in transfer.

**Effectiveness of the proxy KL-divergence loss.** We use proxy KL-divergence loss in the *initialization* function (Sec. 3.1) to compute the importance of units for general knowledge. We are interested in how good the proxy is. We use two kinds of experiments to provide evidences.

**(1)** *Comparing with a sample set of* $D_0$. In some cases, the continual DAP-training users may have the data $D_0$ that was used to pre-train the LM. Then we can just sample a subset from $D_0$ to compute the parameter importance to the general knowledge in the LM. However, since we do not have $D_0$ that was used to pre-train RoBERTa, we use the Wiki data (Merity et al., 2017) as the sample set of $D_0$. We choose it as it is a general dataset with a wide topic coverage and was used to pre-train an LM, and it has a similar size as our domain data (around 700M). We conducted two experiments using the data: **(a) DAS (Wiki+MLM)**, which uses MLM as the loss in the initialization stage to compute the importance of units (to identify the general knowledge) just like any other domains in the continual learning part, and **(b) DAS (Wiki+KL)**, which uses KL-divergence as in the initialization stage just like the proposed proxy method. The results are given in Table 3.

We can see that DAS (Wiki + KL) performs similarly to DAS but outperforms DAS (Wiki + MLM). This indicates that the proposed proxy KL-divergence is more effective. MLM actually adapts the LM to the Wikipedia data, which may not be sufficiently representative of the original data used in pre-training the LM. As a result, it ends up identifying the knowledge that is suitable only for the

| Domain Model | Restaurant MF1 | Acc | ACL MF1 | Acc | AI MF1 | Acc | Phone MF1 | Acc | PubMed MF1 | Camera MF1 | Acc | Average MF1 | Acc | Forget R. MF1 | Acc |
|---|---|---|---|---|---|---|---|---|---|---|---|---|---|---|---|
| DAS (wiki+KL) | **81.25** | **87.89** | **70.89** | **74.87** | 69.68 | 76.86 | 85.98 | **87.78** | 72.03 | 86.69 | 91.44 | 77.75 | 81.81 | -0.50 | -0.27 |
| DAS (wiki+MLM) | 80.22 | 87.12 | 68.12 | 72.92 | 68.55 | 76.06 | 83.50 | 86.11 | 71.94 | 86.02 | 91.15 | 76.39 | 80.88 | 0.54 | 0.40 |
| DAS | 80.34 | 87.16 | 69.36 | 74.01 | **70.93** | **77.46** | **85.99** | 87.70 | **72.80** | **88.16** | **92.30** | **77.93** | **81.91** | **-1.09** | **-0.60** |

Table 3: Results for the Wiki dataset as the sample set of $D_0$ - average of 5 random seeds

| Category | Domain Model | Restaurant MF1 | Acc | ACL MF1 | Acc | AI MF1 | Acc | Phone MF1 | Acc | PubMed MF1 | Camera MF1 | Acc | Average MF1 | Acc | Forget R. MF1 | Acc |
|---|---|---|---|---|---|---|---|---|---|---|---|---|---|---|---|---|
| Non-CL | RoBERTa | 79.81 | 87.00 | 66.11 | 71.26 | 60.98 | 71.85 | 83.75 | 86.08 | 72.38 | 78.82 | 87.03 | 73.64 | 79.27 | — | |
| | DAP-RoBERTa | 80.84 | **87.68** | 68.75 | 73.44 | 68.97 | 75.95 | 82.59 | 85.50 | 72.84 | 84.39 | 89.90 | 76.40 | 80.89 | — | |
| | DAP-Adapter | 80.19 | 87.14 | 68.87 | 72.92 | 60.55 | 71.38 | 82.71 | 85.35 | 71.68 | 83.62 | 89.23 | 74.60 | 79.62 | — | |
| | DAP-Prompt | 79.00 | 86.45 | 66.66 | 71.35 | 61.47 | 72.36 | 84.17 | 86.53 | 73.09 | 85.52 | 90.38 | 74.98 | 80.03 | — | |
| CL DAP-train | DAS (random) | 79.79 | 86.84 | 68.34 | 73.02 | 68.62 | 76.17 | 84.92 | 87.02 | 72.73 | 85.92 | 91.15 | 76.72 | 81.15 | 0.45 | 0.26 |
| | DAS (w/o contrast) | 81.06 | 87.55 | **70.39** | **74.39** | 67.60 | 75.32 | 83.53 | 86.00 | 72.01 | 84.48 | 90.06 | 76.51 | 80.89 | -0.54 | -0.23 |
| | DAS (w/o softmax) | 80.48 | 87.27 | 69.92 | 74.39 | 67.73 | 75.78 | 84.00 | 86.37 | 73.03 | 87.96 | 92.08 | 77.19 | 81.48 | -0.24 | -0.12 |
| | DAS (w/o initialization) | **81.30** | 87.79 | 68.35 | 73.10 | 67.82 | 75.73 | 85.13 | 86.98 | 71.82 | 87.25 | 91.57 | 76.95 | 81.16 | 0.70 | 0.48 |
| | DAS (domain-specific) | 80.95 | **87.68** | 69.18 | 73.21 | 68.92 | 76.27 | 83.89 | 86.26 | 72.46 | 86.74 | 91.53 | 77.02 | 81.23 | -0.07 | 0.08 |
| | DAS | 80.34 | 87.16 | 69.36 | 74.01 | **70.93** | **77.46** | **85.99** | **87.70** | 72.80 | **88.16** | **92.30** | **77.93** | **81.91** | **-1.09** | **-0.60** |

Table 4: Ablation results - averages of 5 random seeds. See *standard deviations* in Appendix E.

Wikipedia data. In contrast, the proposed proxy KL-divergence leverages the random dropout mask and measures the robustness, which is less related to a specific domain and thus reflects the (general) knowledge in the original LM better.

**(2)** *Comparing general knowledge computed from different domain corpora.* Here, we also provide some indirect evidences to show the effectiveness of the proxy method for computing the importance of units to the general knowledge in the LM. We conduct a separate non-CL experiment to compare the attention heads' importance score vectors after applying the proxy using the data from different domains.[12] For each domain $i$, we compare its importance vector with the importance vector of every other domain, and then average the cosine similarities to get the value for domain $i$. We get 0.92 for Restaurant, the same 0.91 for ACL, AI, and Phone, 0.89 for PubMed and 0.92 for Camera. We see that different domains give similar importance values, which indirectly shows that our proxy can approximately identify the common general knowledge.

**Ablation.** We want to know if the proposed (1) initialization (Sec. 3.1), (2) soft-masking, and (3) contrastive learning are helpful. To answer (1), we conduct the ablation **DAS (w/o initialization)**, where we remove the initialization and directly do the continual learning given no consideration to the general knowledge in the LM. To answer (2), we conduct the ablations (1) **DAS (w/o softmask)**, where we remove the soft-masks, and only use contrastive learning based on Eq. 7 (with the second term in the denominator removed); and (2) **DAS (random)** with randomly generated importance scores to do soft-masking and contrastive learning. To answer (3), we conduct two ablations: **(i) DAS (w/o contrast)** where we remove the contrastive loss and only soft-mask according to the importance; **(ii) DAS (domain-specific)** where we contrast domain-specific and learned knowledge (Sec. 3.2). Table 4 shows that the full DAS is the best on average and for most domains, indicating that every component contributes. Additional observations are: (1) DAS's gain is partially from the preserved general knowledge. We can see DAS (w/o initialization) is poorer on average; (2) Soft-masking helps as DAS (w/o softmask) is poorer than DAS. This is reasonable because soft masking can preserve learned domains. Besides, our gradient-based mask is informative as DAS (random) is worse than DAS; (3) Contrastive learning is effective as DAS (w/o contrast) and DAS (domain-specific) are both poorer, indicating the contrastive learning in DAS can help learn good representations

## 5 CONCLUSION

This paper proposed a novel method DAS for the continual DAP-training of an LM. It has three key ideas: (1) Preserving the important previous knowledge by soft-masking units according to their importance to overcome CF and to facilitate knowledge transfer. (2) Using a novel proxy to compute the importance of units to the general knowledge in the LM. (3) Learning complementary representations for knowledge integration. A set of techniques is proposed to achieve them. Extensive experiments showed the effectiveness of DAS. The current approach involves two functions in learning. We will study how to combine them to further improve the results in the future.

---

[12]We use attention heads instead of other units because they are arguably the most important component in a Transformer (Michel et al., 2019; Voita et al., 2019; McCarley et al., 2019).

ACKNOWLEDGEMENTS

The work of Zixuan Ke, Gyuhak Kim, and Bing Liu was supported in part by a research contract from KDDI, a research contract from DARPA (HR001120C0023), and three NSF grants (IIS-1910424, IIS-1838770, and CNS-2225427).

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

## A  DATASETS DETAILS

Table 1 in the main paper has already showed the number of examples in each dataset. Here we provide additional details about the 4 types of end-tasks.

(1) **(Phone, Camera and Restaurant) Aspect Sentiment Classification (ASC)** is defined as follows: given an aspect or product feature (e.g., *picture quality* in a camera review) and a review sentence containing the aspect in a domain or product category (e.g., camera), classify if the sentence expresses a positive, negative, or neutral (no opinion) sentiment or polarity about the aspect (for the Phone and Camera datasets, there are only negative and positive polarities in the data).

(2) **(ACL) Citation intent classification** is defined as follows: given a citing sentence (a sentence contains a citation), classify if the sentence expresses a citation function among "background", "motivation", "uses", "extension" and "comparison or contrast future".

(3) **(AI) Relation classification** is defined as follows: given a within-sentence word sequence span containing a pair of entities, classify if the span expresses a relation among "feature of", "conjunction", "evaluate for", "hyponym of", "used for", "part of" and "compare".

(4) **(PubMed) Chemical-protein interaction classification** is defined as follows: given a span containing a pair of chemical and protein, classify if the span expresses a chemical-protein interaction among "downregulator", "substrate", "indirect-upregulator", "indirect-downregulator", "agnonist", "activator", "product of", "agonist-activator", "inhibitor", "upregulator", "substrate product of", "agonist-inhibitor"and "antagonist".

## B  BASELINE DETAILS

**Non-Continual Learning Baselines**: Each of these baselines builds a separate model for each task independently. It thus has no knowledge transfer or CF.

(1) **Non-DAP-training (RoBERTa)** Liu et al. (2019) uses the original RoBERTa for the end-task fine-tuning without any DAP-training. This is the only one without any DAP-training. All the following baselines use the masked language model loss (MLM) for DAP-training.

(2) **DAP-training using masked language model loss (DAP-RoBERTa)** is the existing DAP-training method in Gururangan et al. (2020). To our knowledge, the existing DAP-training systems are all based on the MLM loss.

(3) **DAP-training using adapter-tuning** Madotto et al. (2020); Houlsby et al. (2019) adds small adapter layers between layers of Transformer for DAP-training. We follow the adapter design in Madotto et al. (2020); Houlsby et al. (2019): An adapter is simply a 2 layers of fully connected network. During DAP-training, the Transformer is fixed, only the added adapters are trainable. The bottleneck size (adapter size) is set to 128. During end-task fine-tuning, both RoBERTa and the adapters are trainable to ensure fair comparison.

(4) **DAP-training using prompt-tuning** Lester et al. (2021) adds a sequence of real vector tokens (called virtual tokens or prompt tokens) to the end of the original sequence. In DAP-training, RoBERTa (the LM) is fixed and only the prompt tokens are trained. In end-task fine-tuning, both LM and the trained prompt are trainable. We initialize 100 tokens and set the learning rate of the prompt token to 0.3 in DAP-training, following the setting in Lester et al. (2021).

**Continual Learning (CL) Baselines.**

(5) **Naive continual learning (NCL)** is a naive extension of Gururangan et al. (2020), which continually/incrementally DAP-trains the LM to learn all domains using MLM loss with no mechanism to deal with CF.

(6) **Continual learning with adapter (NCL-Adapter)** Houlsby et al. (2019) is similar to the adapter based system. The only difference is that the same set of adapters is shared across all domains, rather than using a new adapter for each new domain.

(7) **DEMIX (DEMIX)** Gururangan et al. (2021) is a recent model to adapt pre-trained LM with new domains. It adds a new adapter once a new domain arrives (network expansion is needed) and

initializes the new adapter with the parameters of the previous trained adapter nearest to the new domain data. They use the perplexity on a held-out sample to choose the most probable adapter. For fair comparison, we use the same size as $\{x_n^{\text{sub}}\}$ as the held-out samples.

(8) **Hard attention to overcome forgetting (HAT-Adapter)** Ke et al. (2021c) is derived from HAT Serrà et al. (2018), the state-of-the-art parameter-isolation based method with almost no forgetting. However, HAT requires task id information in end-task fine-tuning (DAS works in *domain-agnostic* manner and does not need the task id information; see Sec. 1). HAT also needs to train an addition task embedding to mask each layer of the network which makes the DAP-training inefficient.

(9) **Continual learning plugin with capsule(BCL)** Ke et al. (2021c) is a continual learning model that can avoid forgetting and encourage knowledge transfer. It is similar to NCL-Adapter. The difference is that its adapters consist of two modules, one is a capsule network (a new capsule is added once a new domain arrives) to encourage transfer and the other one is similar to HAT to avoid forgetting. Similar to HAT, task/domain information is needed in end-task fine-tuning. We replace the backbone network from BERT with RoBERTa for fair comparison.

(10) **Continual learning plugin with contrastive transfer (CLASSIC)** Ke et al. (2021b) is a continual learning model that can avoid forgetting and encourage knowledge transfer via contrasting loss. It is similar to HAT. but 3 additional contrastive loss are used for distillation, knowledge transfer and supervised contrast. Since DAS is working on unsupervised data, we remove the supervised contrastive loss. Similar to HAT, task information is needed in end-task fine-tuning. We replace the backbone network from BERT with RoBERTa for fair comparison.

(11) **Knowledge distillation (KD)** Hinton et al. (2015) minimizes the representational deviation between the learned representation and the new representation in DAP-training. We compute the KL divergence between the representations (the output before the masked language model prediction head) of each token of the previous DAP-trained LM and current LM as the distillation loss.

(12) **EWC** Buzzega et al. (2020) is a popular regularization-based method which adopts elastic weights consolidation to add $L_2$ regularization to parameter changes.

(13) **DER++** Buzzega et al. (2020) is a recent replay method using distillation to regularize the new task training. We store 16.4K tokens for each learned domain as the memory, which is the largest memory we can use for the system to run.

(14) **HAT** Serrà et al. (2018) is used in the Transformer layers (including self-attention, intermediate and output layers) rather than the added adapter layers. Additional task embedding and task information for end-task fine-tuning are needed.

## C  IMPLEMENTATION DETAILS

**Architecture.** We adopt RoBERTaBASE as our backbone LM. A masked language model head is applied for DAP-training. The end-task fine-tuning of RoBERTa follows the standard practice. For the three ASC tasks (see Table 1), we adopt the ASC formulation in Xu et al. (2019), where the aspect (e.g., "*sound*") and review sentence (e.g., "*The sound is great*") are concatenated via .

**Hyperparameters.** Unless otherwise stated, the same hyper-parameters are used in all experiments. The maximum input length is set to 164 which is sufficient for all datasets. Adam optimizer is used for both DAP-training and end-task fine-tuning. The max sequence length is also set to 164.

**DAP-training.** The learning rate is set to 1e-4 and batch size to 256. We train 2.5K steps for each domain, roughly a full pass through the domain data, following Gururangan et al. (2020); Xu et al. (2019). The subset of data $\{x_n^{\text{sub}}\}$ for computing $\mathcal{L}^{\text{impt}}$ to determine head importance in Secs. 3.1 and 3.3 is set to 1.64 Million tokens, which is sufficient in our experiments. $\lambda$ in Eq. 8 is set to 1 and $\tau$ in Eq. 7 is set to 0.05.

**End-task fine-tuning.** The learning rate is set to 1e-5 and batch size to 16. We train on end-task fine-tuning datasets for 5 epochs for Restaurant; 10 epochs for ACL, AI and PubMed; and 15 epochs for Phone and Camera. We simply take the results for the last epoch, assuming no validation sets. We empirically found that the above number of epochs gives us stable and convergence results.

| Random Sequence Order | NCL MF1 | NCL Acc | KD MF1 | KD Acc | DAS MF1 | DAS Acc |
|---|---|---|---|---|---|---|
| Restaurant → ACL → AI → Phone → PubMed → Camera | 76.36 | 80.77 | 75.17 | 80.06 | **77.93** | **81.91** |
| Phone → AI → PubMed → Camera → Restaurant → ACL | 76.06 | 80.69 | 75.49 | 80.39 | **76.90** | **81.10** |
| PubMed → Camera → ACL → Restaurant → AI → Phone | 76.49 | 80.84 | 75.80 | 80.51 | **76.86** | **81.09** |
| Camera → ACL → Phone → Restaurant → PubMed → AI | 76.28 | 80.83 | 74.67 | 79.91 | **77.52** | **81.65** |
| AI → PubMed → Camera → Phone → ACL → Restaurant | 76.17 | 80.68 | 75.32 | 80.45 | **78.18** | **82.10** |

Table 5: DAS performance averaged over all domains after the final DAP-trained (averaged over 5 random seeds).

| Category | Domain Model | Restaurant MF1 | Restaurant Acc | ACL MF1 | ACL Acc | AI MF1 | AI Acc | Phone MF1 | Phone Acc | PubMed MF1 | Camera MF1 | Camera Acc | Average MF1 | Average Acc |
|---|---|---|---|---|---|---|---|---|---|---|---|---|---|---|
| | Pool | ±0.0070 | ±0.0032 | ±0.0177 | ± 0.0103 | ±0.0137 | ±0.0087 | ±0.0190 | ±0.0142 | ±0.0088 | ±0.0345 | ±0.0209 | ±0.0127 | ±0.0085 |
| Non-CL | RoBERTa | ±0.0117 | ±0.0049 | ±0.0192 | ±0.0096 | ±0.0646 | ±0.0347 | ±0.0210 | ±0.0154 | ±0.0071 | ±0.0403 | ±0.0179 | ±0.0119 | ±0.0070 |
| | DAP-RoBERTa | ±0.0096 | ±0.0056 | ±0.0218 | ±0.0118 | ±0.0117 | ±0.0086 | ±0.0165 | ±0.0103 | ±0.0035 | ±0.0479 | ±0.0298 | ±0.0118 | ±0.0075 |
| | DAP-Adapter | ±0.0102 | ±0.0068 | ±0.0142 | ±0.0099 | ±0.0551 | ±0.0288 | ±0.0265 | ±0.0181 | ±0.0055 | ±0.0165 | ±0.0110 | ±0.0132 | ±0.0087 |
| | DAP-Prompt | ±0.0060 | ±0.0035 | ±0.0068 | ±0.0108 | ±0.0301 | ±0.0124 | ±0.0126 | ±0.0087 | ±0.0028 | ±0.0243 | ±0.0138 | ±0.0049 | ±0.0019 |
| | NCL | ±0.0064 | ±0.0035 | ±0.0168 | ±0.0084 | ±0.0164 | ±0.0099 | ±0.0126 | ±0.0104 | ±0.0073 | ±0.0449 | ±0.0247 | ±0.0116 | ±0.0073 |
| | NCL-Adapter | ±0.0090 | ±0.0060 | ±0.0063 | ±0.0065 | ±0.0835 | ±0.0405 | ±0.0196 | ±0.0124 | ±0.0086 | ±0.0312 | ±0.0152 | ±0.0117 | ±0.0058 |
| | DEMIX | ±0.0065 | ±0.0029 | ±0.0118 | ±0.0094 | ±0.0376 | ±0.0218 | ±0.0731 | ±0.0428 | ±0.0069 | ±0.0099 | ±0.0071 | ±0.0121 | ±0.0064 |
| | BCL | ±0.0106 | ±0.0059 | ±0.0050 | ±0.0054 | ±0.0433 | ±0.0229 | ±0.0191 | ±0.0130 | ±0.0069 | ±0.0290 | ±0.0164 | ±0.0097 | ±0.0055 |
| | CLAASSIC | ±0.0071 | ±0.0039 | ±0.0337 | ±0.0171 | ±0.0227 | ±0.0084 | ±0.0187 | ±0.0124 | ±0.0085 | ±0.0140 | ±0.0094 | ±0.0114 | ±0.0065 |
| CL DAP-train | KD | ±0.0352 | ±0.0197 | ±0.0096 | ±0.0107 | ±0.0164 | ±0.0088 | ±0.0149 | ±0.0115 | ±0.0075 | ±0.0277 | ±0.0128 | ±0.0072 | ±0.0042 |
| | EWC | ±0.0161 | ±0.0085 | ±0.0136 | ±0.0076 | ±0.0178 | ±0.0089 | ±0.0205 | ±0.0140 | ±0.0069 | ±0.0725 | ±0.0424 | ±0.0172 | ±0.0098 |
| | DER++ | ±0.0081 | ±0.0042 | ±0.0156 | ±0.0089 | ±0.0402 | ±0.0160 | ±0.0402 | ±0.0272 | ±0.0090 | ±0.0367 | ±0.0215 | ±0.0158 | ±0.0088 |
| | HAT | ±0.0182 | ±0.0091 | ±0.0271 | ±0.0206 | ±0.0369 | ±0.0126 | ±0.0834 | ±0.0474 | ±0.0038 | ±0.1082 | ±0.0408 | ±0.0323 | ±0.0155 |
| | HAT-All | ±0.0257 | ±0.0140 | ±0.0643 | ±0.0273 | ±0.1355 | ±0.0991 | ±0.0428 | ±0.0217 | ±0.0125 | ±0.0526 | ±0.0163 | ±0.0175 | ±0.0145 |
| | HAT-Adapter | ±0.0093 | ±0.0061 | ±0.0048 | ±0.0053 | ±0.0289 | ±0.0168 | ±0.0277 | ±0.0195 | ±0.0037 | ±0.0760 | ±0.0370 | ±0.0129 | ±0.0074 |
| | DAS | ±0.0090 | ±0.0063 | ±0.0186 | ±0.0103 | ±0.0142 | ±0.0086 | ±0.0160 | ±0.0135 | ±0.0067 | ±0.0289 | ±0.0154 | ±0.0099 | ±0.0060 |

Table 6: Standard deviations of the corresponding metrics of the proposed DAS model and the baselines

# D DAP-TRAINING IN DIFFERENT ORDERS

Table 2 in the main paper reported the results for the order Restaurant → ACL → AI → Phone → PubMed → Camera. We now look at how the order affects the results. Due to the computation intensive nature of DAP-training, we only report the best baseline (NCL) and DAS results with different domain orders. Table 5 shows NCL and DAS's results of 5 different orders. We can see DAS is always better than NCL, demonstrating the effectiveness of DAS.

# E STANDARD DEVIATIONS

Table 6 reports the standard deviations of the corresponding results in Table 2 (in the main paper) of DAS and the considered baselines over 5 runs with random seeds. We can see the results of DAS are stable. Some baselines (e.g., RoBERTa in AI, DAP-RoBERTa in Camera) can have quite large standard deviations.

Table 7 reports the standard deviations of the corresponding results in Table 4 (in the main paper) of DAS and the considered baselines over 5 runs with random seeds. We can see the results of DAS and its variants are stable.

| Category | Domain Model | Restaurant MF1 | Restaurant Acc | ACL MF1 | ACL Acc | AI MF1 | AI Acc | Phone MF1 | Phone Acc | PubMed MF1 | Camera MF1 | Camera Acc | Average MF1 | Average Acc |
|---|---|---|---|---|---|---|---|---|---|---|---|---|---|---|
| | RoBERTa | ±0.0117 | ±0.0049 | ±0.0192 | ±0.0096 | ±0.0646 | ±0.0347 | ±0.0210 | ±0.0154 | ±0.0071 | ±0.0403 | ±0.0179 | ±0.0119 | ±0.0070 |
| Non-CL | DAP-RoBERTa | ±0.0096 | ±0.0056 | ±0.0218 | ±0.0118 | ±0.0117 | ±0.0086 | ±0.0165 | ±0.0103 | ±0.0035 | ±0.0479 | ±0.0298 | ±0.0118 | ±0.0075 |
| | DAP-Adapter | ±0.0102 | ±0.0068 | ±0.0142 | ±0.0099 | ±0.0551 | ±0.0288 | ±0.0265 | ±0.0181 | ±0.0055 | ±0.0165 | ±0.0110 | ±0.0132 | ±0.0087 |
| | DAP-Prompt | ±0.0060 | ±0.0035 | ±0.0068 | ±0.0108 | ±0.0301 | ±0.0124 | ±0.0126 | ±0.0087 | ±0.0028 | ±0.0243 | ±0.0138 | ±0.0049 | ±0.0019 |
| | DAS (random) | ±0.0074 | ±0.0055 | ±0.0110 | ±0.0102 | ±0.0201 | ±0.0112 | ±0.0184 | ±0.0128 | ±0.0042 | ±0.0483 | ±0.0247 | ±0.0119 | ±0.0067 |
| CL DAP-train | DAS (w/o contrast) | ±0.0104 | ±0.0055 | ±0.0090 | ±0.0063 | ±0.0205 | ±0.0124 | ±0.0321 | ±0.0216 | ±0.0037 | ±0.0527 | ±0.0286 | ±0.0119 | ±0.0073 |
| | DAS (w/o softmask) | ±0.0064 | ±0.0046 | ±0.0121 | ±0.0088 | ±0.0193 | ±0.0113 | ±0.0245 | ±0.0175 | ±0.0096 | ±0.0322 | ±0.0183 | ±0.0104 | ±0.0059 |
| | DAS (w/o initialization) | ±0.0124 | ±0.0075 | ±0.0054 | ±0.0048 | ±0.0134 | ±0.0078 | ±0.0135 | ±0.0104 | ±0.0118 | ±0.0460 | ±0.0261 | ±0.0093 | ±0.0058 |
| | DAS (domain-specific) | ±0.0067 | ±0.0045 | ±0.0151 | ±0.0127 | ±0.0192 | ±0.0129 | ±0.0277 | ±0.0182 | ±0.0061 | ±0.0419 | ±0.0226 | ±0.0120 | ±0.0077 |
| | DAS | ±0.0090 | ±0.0063 | ±0.0186 | ±0.0103 | ±0.0142 | ±0.0086 | ±0.0160 | ±0.0135 | ±0.0067 | ±0.0289 | ±0.0154 | ±0.0099 | ±0.0060 |

Table 7: Standard deviations of the corresponding metrics of the proposed DAS model and the ablation

