# OpenReview forum: "Continual Pre-training of Language Models"
_ICLR.cc/2023/Conference — ICLR 2023 poster_

### Official Review · Reviewer_ApDZ · 2022-10-18

**Confidence:** 4
**Correctness:** 4
**Technical Novelty And Significance:** 4
**Empirical Novelty And Significance:** 3
**Recommendation:** 8

**Clarity, Quality, Novelty And Reproducibility:**

The paper is clearly written and introduces several new and well-motivated ideas.

**Strength And Weaknesses:**

The paper provides a principled approach to mitigating the forgetting problem when learning over time using sequential data sets from differing domains.  The solution is elegant and practical.

**Summary Of The Paper:**

This paper addresses the problem of sequentially pretraining a neural language model on a sequence of domain specific training sets whilst preserving the original pretaining and avoiding catastrophic forgetting.  The basic idea is to maintain a soft mask over critical neurons (in this case the attention heads) such that backward error propagation is scaled down for paths which encode most of the existing knowledge.  The mask weights are learned by introducing a temporary gating function and using the normalised gate weights as the mask.  Mask weights are updated for each new dataset by taking the max of the mask for the previous iteration and the current.  In addition the normal LM Mask loss is augmented by a contrastive loss designed to encourage the LM to acquire knowledge from the new data that is not already in the knowledge extracted from previous domains.  In the experimental evaluation, the proposed system is extensively tested against a number of existing continual learning paradigms with impressive results.  Over a sequence of 6 datasets, the forgetting rate for the proposed system is negative showing that it not only prevented forgetting but it also facilitated knowledge transfer from one domain to another.  Ablation tests show that each component of the proposed system makes a positive contribution.

**Summary Of The Review:**

Overall this is a very nice paper which introduces a novel approach to a problem which has significant practical importance for real world applications.  The methods here are presented in the context of large LMs, but they should be applicable to a wide range of neural network systems. The work is clearly presented and the experimental work is thorough and convincing.

---

> ### Author Response · Authors · 2022-11-18
> **Response to Reviewer ApDZ**
>
> Thank you very much for appreciating our work.

---

### Official Review · Reviewer_8TT6 · 2022-10-23

**Confidence:** 2
**Clarity, Quality, Novelty And Reproducibility:** 1. I cannot understand most of the te…
**Correctness:** 3
**Technical Novelty And Significance:** 2
**Empirical Novelty And Significance:** 3
**Recommendation:** 3

**Strength And Weaknesses:**

1. the general ideal of training different important parameters for each domain, and preserves the previous domains is pretty interesting


**Summary Of The Paper:**

This paper proposes to continue adapting the language model to new domains without forgetting the previous domains. The main method first calculates the important neurons based on general language knowledge. Then it uses a KL divergence loss to to continued adaptation to other tasks and domains.

**Summary Of The Review:**

The main issue with the paper is that the writing of the technical methods are very unclear, so it's difficult for me to understand or assess it. It might not be suitable for publishing at its current stage because the writing is not clear enough for people to replicate the method.

---

> ### Author Response · Authors · 2022-11-18
> **Response to Reviewer 8TT6**
>
> > what the motivation behind proxy KL-divergence loss is? One could sample some pretraining data, and use the MLM objective to calculate eqn 3).
>
> This may be a misunderstanding. We are working on continual post-training, where we start continual learning from a pre-trained LM. This is different from continual pre-training, where the system starts continual learning from a randomly initialized LM. In continual pre-training, one builds the LM from scratch and can access the data that pre-trains the LM. In continual post-training, which is done by an end-user, s/he only uses the LM and builds on it, and thus does not normally have access to the pre-training data of the LM but only his/her own domain data. Even if the company that trained the LM can open their data, it will be too large for an end-user to download or to use.
> >  I_{l, i}^{gen} to indicate the importance of the units, but this notation is not used anywhere else. Is that basically just the gradient, which is delta_{gl}? There are many notations in this paragraph that are not used anywhere else
>
> We believe all used notations are defined. In the paragraph before “proxy KL-divergence”, the first sentence, writes “the importance of units to the general knowledge in the LM ($I_{l, i}^{gen}$)” and also the first paragraph in 3.2: “the importance of units to the general knowledge ($I_{l, i}^{gen}$)”. We believe it is clear what this notation means. The importance of units for general knowledge is not simply the gradient of the gate vector ($\nabla_{g_l}$). Instead, it is computed from Eq. 3. and further normalized by Eq. 5 before being used to do soft-masking (In the paragraph after Eq. 6, it states "$I_l^{(\leq0)}$ is $I_{l}^{gen}$"
> > what "no label in post-training" mean
>
> It just means that the data is unlabeled as it is well understood that post-training is applied to an unlabeled domain corpus. The KL-divergence loss also does not need any label. We have revised this part to make it clearer.
> > what EMax means
>
> It means element-wise max
> >  It might be helpful to just use o_prev and o_full
>
> We actually considered this but decided not to because we do not want to make equation 11 too complicated. The relationship between Eq. 9-10 and Eq. 11 is stated in the paragraph before Eq. 11
> > notation L_{constraint}, which is not used anywhere else
>
> We define L_contrast in Eq. 11 and use it in Eq. 12.
> > not clear whether the method would work well on multiple domains
>
>  This is a misunderstanding. All our experiments involve multiple domains and they are learned incrementally one by one.
> > The main issue with the paper is that the writing...not clear enough for people to replicate the method.
>
> We notice some misunderstandings here and there, which may be because of our writing. We further update the introduction for better understanding. Replication is easy because we have provided the code. We hope that you can change your mind as we believe that we have presented a novel idea, which you can also see from the review of Reviewer ApDZ. We notice that your original Confidence score was 1. We assume that our paper or continual learning is not in your area of expertise, which perhaps makes it harder to read our paper. This paper is also quite involved with a lot of details about continual learning and language models.

---

### Official Review · Reviewer_YGnc · 2022-10-24

**Confidence:** 5
**Correctness:** 3
**Technical Novelty And Significance:** 2
**Empirical Novelty And Significance:** 3
**Recommendation:** 8

**Clarity, Quality, Novelty And Reproducibility:**

How about just pool all data and train only once? What would be that performance, obviously continual updating the model with newly acquired data would not be feasible with this
approach, but it is necessary to know how much we lose by doing continual learning.

I feel that too many acronyms are used, such as CF and TIL, please consider eliminating unnecessary ones. And in the same vein, why the first TIL occurrence does not have a reference?

Introduction: About properties required, to me it seems like property (1) and (2) are same. Both essentially talk about catastrophic forgetting. Please clarify.
Footnote 2: I would say instead that definition of catastrophic forgetting is that training on new domains causes performance on previous domain to degrade. So to strengten that statement.

Section 2:
Is difference between ELLE and the proposed method only that you start with pretrained model and ELLE starts from scratch? If so, I guess your novelty is then very limited. Please correct this statement.

I would contest these statements: "(1) we compute importance for soft masking, not for pruning, and (2) post-training works on unlabeled data, while the above approaches work with labeled data."
First of all, computing importance for soft masking and pruning can be considered to be the same. If for pruning you need to compute a score each node and then apply some thresholding, well then you can apply same scoring technique for your soft masking. Same goes for labeled vs unlabeled case, in both cases you have loss function that is being optimized. There is no
principal difference why method developed for labeled data could not be used for unlabeled data.

In don't understand why contrastive learning is mentioned in this context.

Section 3: How is KL in (4) computed? What exactly is the input to it? Mathematical details are missing here.
In proxy loss you take subset of the _current_ domain data, how you can claim that gradient of the KL then represents general knowledge?

Equation (5), I am confused, where is mean and Var taken of, over all I_l?  Please clarify.

Equation (6), what is EMax?

**Strength And Weaknesses:**

Strengths:
- Good empirical results in a very relevant task (self-supervised LM) with a number of different target domains.
- Approach seems to make intuitive sense.

Weaknesses:
- Writeup is not finalized (see explanation in the clarity box)
- Theoretical grounding is missing. Final result appears to be empirical searched solution to a practical problem, while idea itself is sensible. In my opinion, authors should try to answer theoretically that in some restricted case the proposed method would get the "correct" answer. Other way would be to show that it cannot completely fail.

**Summary Of The Paper:**

In this paper authors propose to improve continual post-training by the way of computing saliency of each unit or layer in transformer model. Average saliency is then used as a weight in the backward pass. Idea appears to be first time used in relation to self-supervised loss functions such as MLM as is the case here. Authors overcome multiple technical challenges in their system and the final experimental results are quite convincing.

**Summary Of The Review:**

Idea does make intuitive sense, but it is my opinion that paper at the current state is not of sufficient quality.

After rebuttal:

I find authors rebuttal and changes to be substantial enough to change my score from 5 -> 8.

---

> ### Author Response · Authors · 2022-11-18
> **Response to Reviewer YGnc (Part 1)**
>
> > Theoretical grounding is missing
>
> Regarding theoretical analysis, we actually thought about it carefully, but have not found a good approach at the moment as our work involves both knowledge transfer and catastrophic forgetting. Traditionally, there is limited theoretical work on knowledge transfer except about generation bounds. For example, there is a recent theoretical work on knowledge transfer [a], but it mainly says that a sequence of related or similar tasks will result in knowledge transfer and the larger the number of tasks, the better the bounds. However, this provides little guidance on how to do knowledge transfer, not to mention that our work also considers dissimilar tasks. Furthermore, [a] does not consider catastrophic forgetting. The combination of knowledge transfer and forgetting makes it harder to prove anything, but this is a very important topic for future study. Maybe some more empirical work will enable a better understanding of the problem before formulating a theoretical framework.
>
> [a] Benavides-Prado D, Riddle P. A Theory for Knowledge Transfer in Continual Learning. Collas-2022.
> > How about just pool all data and train only once?
>
> Since this is not a continual learning baseline, we did not include it. We have conducted a new experiment and reported the results of this method (called “Pool”) in Table 2. We can see that CPS on average actually performs better than Pool. This could be because some domains are quite different (e.g. camera reviews and ACL papers) and there could be some negative interference in POOL. CPS is able to better avoid this because in learning each task, it can leverage the preserved general knowledge and learn in its own space with the previous task knowledge protected by soft-masks.
> > I feel that too many acronyms are used
>
> Sorry for confusing you. We summarize the acronyms we use as followings: (1) LM refers to Language Model; (2) CL refers to continual learning; (3) CF refers to catastrophic forgetting; (4) KT refers to knowledge transfer; (5) EMax refers to element-wise max in Eq. 6, and (6) SM refers to soft-making in Eq.12. We also use units to refer to both attention heads and neurons in the Transformer. We have removed the acronym PI (parameter-isolation) and TIL (Task-incremental learning) , which are unnecessary.
> > Introduction: About properties required, to me it seems like property (1) and (2) are same. Footnote 2: I would say instead that definition of catastrophic forgetting..
>
> It is true that (1) and (2) are both about forgetting, but this is a subtle difference.  For (1), it is about the pre-trained language model (LM), for which we only want to preserve the general language knowledge in the LM and the LM still can be adapted/changed to suit individual domains in continual learning. The method used to detect and to preserve the knowledge (see Sec. 3.1 the initialization stage) is different from that for (2) (see Sec. 3.2 the continual learning stage). (2) is the traditional forgetting about individual domains/tasks. About the definition of catastrophic forgetting, we followed your suggestion and changed it.
> > Is difference between ELLE and the proposed method only that you start with pretrained model and ELLE starts from scratch
>
> Sorry, we should have said more. Apart from what we already wrote, ELLE is a combination of network expansion and replay. For each task, it expands the network and the training uses the traditional replay method. However, it saves a huge amount of data (1G per domain) for replay. It also assumes that it has the data used to pre-train the original language model, which is not realistic for end-users. These have been added to the updated version in the related work section in page 3.
> > I would contest these statements (1) we compute importance for soft masking, not for pruning...
>
> What we meant was that we cannot directly use the original pruning technique (Eq. 3) because CPS and pruning are for different purposes and using different data. More specifically, in the initialization stage (Sec. 3.1), we need to compute the importance of units for general knowledge without accessing the pre-training data. The existing pruning methods cannot achieve this and thus we propose to use the proxy KL-divergence. In the continual learning stage (Sec. 3.2), since we have the domain data, we can adopt the pruning technique (yes, the idea of importance computation is the same here) but we do not use a threshold to prune the network because we want to accumulate knowledge rather than to compress the LM. Also, the existing pruning method is for traditional classification with labeled training data but our data is unlabeled domain corpus. We had to change the loss to mask language model loss (Sec. 3.3) (here we convert the unsupervised objective to self-supervised objective, making it similar to the supervised objective in pruning). We have made it clearer in the revised paper (the “neural network pruning” section in Sec. 2).

---

> > ### Author Response · Authors · 2022-11-18
> > **Response to Reviewer YGnc (Part 2)**
> >
> > > why contrastive learning is mentioned in related work
> >
> > We mentioned it in Section 2 (the last sentence) because we need to use it in our algorithm for knowledge integration (or complementary effect) later.
> > > How is KL in (4) computed? you take subset of the current domain data, how you can claim that gradient of the KL then represents general knowledge
> >
> > The KL we use is the standard KL-divergence, i.e., KL(a,b) = b * (log(b) - a). The inputs to eq. 4 are two representations from the same sample. We get these two representations by feeding the same sample to the Transformer twice. This gives us two different representations because of the dropout mask in the standard Transformer (These are mentioned right after Eq. 4).
> >
> > The second part of your question involves two sub-questions if we understand you correctly, (1) one is why a subset is good enough to compute the importance of units for general knowledge compared to a full set. (2) The other one is how good the proxy is.
> >
> > For (1), we did not use the full set because the post-training data is supposed to be very large in practice. We have conducted a new experiment using the full set to compute the importance. The full set results (average MF1: 77.45; Acc.: 81.54), are very similar to the subset’s counterpart (average MF1: 77.93; Acc.: 81.91), indicating the subset is good enough for important computation.
> >
> > For (2), we discussed this problem in the “Effectiveness of the proxy KL-divergence loss” sub-section in Sec. 4.2. The main idea is that although it is hard to assess the quality of the proxy since we don’t have the pre-training data, some indirect evidence shows the effectiveness of the proxy for computing the importance of units to the general knowledge in the LM.
> > >  where is mean and Var taken of, over all I_l?
> >
> > Yes, they are computed from the importance of all units from $I_l$. We updated the equation to make it clearer.
> > > what is EMax
> >
> > It means element-wise max. We have made it clearer before Eq. 6.
> > >Idea does make intuitive sense, but it is my opinion that paper at the current state is not of sufficient quality.
> >
> > We further update the introduction for better understanding. As you can see, your comments (mainly minor misunderstandings due to our writing) are easily addressed, which we have done. We hope that you can change your mind as we believe that we have presented a novel idea, which you can also see from the review of Reviewer ApDZ.

---

> > > ### Comment · Reviewer_YGnc · 2022-11-29
> > > **I am quite happy with the rebuttal and the revisioning**
> > >
> > > I am quite happy about the current state of the paper based on the expert revisioning and rebuttal by the authors.

---

> ### Author Response · Authors · 2022-11-28
> **A gentle reminder for Reviewer YGnc**
>
> Dear Reviewer YGnc,
>
> We are writing to check whether our response has addressed your concerns. If you have any further questions, we will be very happy to answer. In case you didn’t notice, reviewer dAfE and us had an extensive discussion and s/he is now happy with our work. Some of his/her questions and our answers may be helpful to clear your doubts as well.
>
> Thanks,
> Authors
>
> PS: We understand Reviewer 8TT6 has not responded as his/her expertise is not in this area (the confidence score is 2, and was 1 originally).

---

### Official Review · Reviewer_dAfE · 2022-11-17

**Confidence:** 5
**Correctness:** 4
**Technical Novelty And Significance:** 3
**Empirical Novelty And Significance:** Not applicable
**Recommendation:** 8

**Clarity, Quality, Novelty And Reproducibility:**

- Clarity: The clarity can be significantly improved, including especially in Section 3. Also, more descriptions of the experiments should be included in Section 4.
- Quality: The quality can be improved, regarding the experiment setting, baseline comparisons, and the clarity of the paper.
- Originality: The training objectives of the CPS are actually quite novel but unintuitive and unconvincing.
- Reproducibility: Good, the authors provide the source code.


**Strength And Weaknesses:**

Update after discussing with the authors
===

After all the supplementary experiments added during the discussion period, I find this paper and the proposed method to be solid and convincing.
I highly appreciate the efforts done by the authors and I find this process to be very beneficial.
**All the weaknesses are perfectly addressed.**
I am thus raising the score from 3 to 8.

-------------

Strengths
===
1. Propose a method that shows empirical performance gain in language model post-training
2. The paper contains a lot of baseline methods for comparison.

Weaknesses
===
1. The intuitions of the proposed method are not very convincing. These include the loss function in the initialization and continual learning, and the importance computation, which are the core of CPS.
   - I do not see why we need to assume "general domain datasets are not accessible to users of the LM". Those general domain corpora should be very easy to obtain in realistic, and that is why they are used for self-supervised pre-training.
   - I cannot agree with why using the domain-specific dataset and the loss function in Equation (4) is for learning general domain knowledge.
   - I do not understand why contrastive learning is used here. Why would we want the representation of full knowledge and previously learned knowledge to be pushed away in the representation space, and how does this connected with knowledge "integration"?

2. Some previous works and baselines should be investigated, this includes
   - "Jin et al. (2021) compared several CL methods for continual post-training", as stated in Section 2. In fact, the scenario discussed in this paper is highly overlapped with Jin et al. (2021). In this case, the methods mentioned by Jin et al. (2021) should be compared. However, this paper misses a comparison between most knowledge distillation-based methods in Jin et al. (2021), which are exactly the methods that achieve better performance. There are 5 different KD methods in Jin et al. (2021) but there is only one KD method in this paper.
    -  I also think LoRA [1] should be used as a baseline.
    - Why is AdapterFusion not compared here? One can train domain-specific adapters using unlabeled data (MLM), and use the domain-specific downstream task dataset to learn the parameter of AdapterFusion and the downstream task's parameter.

3. It is unclear whether the soft-masking based on importance is really very critical to the proposed method.
    - I am curious whether CPS is successful because when updating the model, the soft-masking is calculated based on the importance, or it is just because we are suppressing the update of some proportion of the weights, and where the weights whose gradient update is soft-masked is not important. Gating the gradient during backpropagation sounds like some kind of regularization, and I would like to know how the regularization is placed is really important. To be specific, I would like to see an ablation study that constructs a random importance score $I_{l}^{(k)}$ that samples from a fixed probability distribution on $[0,1]$, and see whether the performance degrades or not.

4. The paper is not very clear, including:
    - What does $f_{LM}(o_l^{prev})$ mean? What does "plug it into the whole language model" means here? Is this simply using $I_{l}^{(\leq t-1)}$ in the transformer model?
    - How is NCL (Adapter) done? I understand that this is described in the appendix, but I find that most experiment details related to the baseline methods are placed in the appendix, making the paper's main content not very well self-contained. An additional question is why not add a new adapter in a new domain as in [2]? This is like DEMIX but different initialization method.
    - The intuitions for the methods are not very convincing, making me find those parts confusing and not very clear.
    - How is the model fine-tuned on the end tasks in CL baselines? Full model fine-tuning with a classifier head or using parameter-efficient fine-tuning methods such as adapter or LoRA? Does fine-tuning methods change the conclusion of what CL method is better?

5. Lack of comparison when the learning order of domains is permuted. In Table 4, only the performance of CPS and NCL are compared, but this Table should have compared other CL methods.

[1] Hu, Edward J., et al. "LoRA: Low-Rank Adaptation of Large Language Models." International Conference on Learning Representations. 2021.
[2] Jang, Joel, et al. "Towards Continual Knowledge Learning of Language Models." International Conference on Learning Representations. 2021.

**Summary Of The Paper:**

This paper proposes Continual Post-training of LMs with Soft-masking (CPS) that adapts the pre-trained language models on different domains in a continual-learning way. CPS is designed such the language model can maximally preserve the knowledge learned in the general domain and previously-seen knowledge-specific domain. This is done by calculating the importance of submodules in the transformer models and using the importance score to calculate the soft-mask. The soft-mask is used to mask out gradients such that important units for the previous domains will not be changed too much. The experiment results of continual post-training on six different domains show that the proposed method, CPS, outperforms other continual learning and non-continual learning baselines.

**Summary Of The Review:**

Update after discussing with the authors
===
I think this paper provides a solid and convincing method for language model post-training. While there may be minor issues with the writing, the method itself is good. I think this paper should be accepted.

-------------
I think the paper requires significant improvements, including experiment setting and writing. I do not think the paper is ready to be accepted to ICLR.

---

> ### Author Response · Authors · 2022-11-18
> **Response to Reviewer dAfE (Part 1)**
>
> > why assume "general domain datasets are not accessible to users of the LM"
>
> This may be a misunderstanding. We are working on continual post-training, where we start continual learning from a pre-trained LM. This is different from continual pre-training, where the system starts continual learning from a randomly initialized LM. In continual pre-training, one builds the LM from scratch and can access the data that pre-trains the LM. In continual post-training, which is done by an end-user, s/he only uses the LM and builds on it, and thus does not normally have access to the pre-training data of the LM but only his/her own domain data. Even if the company that trained the LM can open their data, it will be too large for an end-user to download or to use.
>
> > I cannot agree with why using the domain-specific dataset and the loss function in Equation (4)
>
> We respectfully ask you why you cannot agree. Our intuitions are given from Eq. 3 to the end of Sec. 3.1. The results in Sec. 4, including overall results, ablation results, and further analysis on the effectiveness of KL-divergence all indicate Eq. 4 is very effective.
>
> > why contrastive learning is used here
>
> The reason is given in Sec. 3.2 (3): “to encourage the current domain representation to learn knowledge that is not already in the learned knowledge in previous domains and yet related to and integrated with the learned knowledge.” Specifically, integration means we want to combine the current domain knowledge and the previously learned knowledge without destroying any of them. Ideally, they should be complementary to each other. In our case, the full knowledge contains all previously learned knowledge. When pushing it away, the model is forced to learn the current domain knowledge which was originally mixed with the previous knowledge. This will not destroy the previously learned knowledge protected by soft-masking. Since we still have the $L_{MLM+SM}$ (Eq. 12), the two types of knowledge tend to be different and do well together for the new domain (i.e., complementary).
>
> >  this paper misses a comparison between most knowledge distillation-based methods in Jin et al. (2021), there is only one KD method in this paper.
>
> we indeed noticed Jin et al, (2021) and cited in the related work. To answer your questions: (1) we have two KD based methods, one is KD, corresponding to the rep-KD in Jin et al, 2021. The other one is DER++, which corresponds to logit-KD and further with replay data (ER in Jin et al, 2021). These two are the most effective KD according to Jin’s work. You can see from Table 3 that CPS outperforms all these. (2) The issue with Jin’s work is that none of their compared baselines are from recent state-of-the-art. Further, they are all adapted from image classification. None of them are designed specifically for NLP tasks. CPS addresses these in experiment design: We include recent state-of-the-art include HAT, DER++ and DEMIX. We also include some NLP-specific baselines including BCL and CLASSIC. We believe our experiments are more complete and solid.
>
> >  LoRA [1] should be used as a baseline
>
> We have included both prompt and adapter, which are representatives of related approaches. Adapter is one of the most effective parameter-efficient techniques. In both [a] and [b], we can see the adapter consistently outperforms other parameter-efficient techniques including prefix-tuning and LoRA. We have discussed and cited this paper in footnote 11.
>
> [a]: Mao et al., UniPELT: A Unified Framework for Parameter-Efficient Language Model Tuning, ACL 2022
>
> [b]: He et al., Towards a Unified View of Parameter-Efficient Transfer Learning, ICLR 2022
>
> > Why is AdapterFusion not compared here
>
> This paper works on continual post-training. Our focus is on how to achieve CF prevention and KT in the post-training stage. How to better fuse the knowledge in downstreaming tasks (e.g., adapterfusion) is out-of-the-scope of this paper. We have discussed and cited this paper in footnote 11.
>
> > To be specific, I would like to see an ablation study that constructs a random importance score
>
> We added the CPS (random) in Table 3. We can see it is worse than CPS and suffers from forgetting. This indicates our importance-based masking is effective.
>
> > What does "plug it into the whole language model" means
>
>  Since $O_l^{prev}$ only refers to the output of layer $l$. Plugging it into the LM means using it for each layer of the LM. In brief, this means that we use the LM with soft-masking. In contrast, in Eq. 10, we use the LM without any masking.

---

> > ### Author Response · Authors · 2022-11-18
> > **Response to Reviewer dAfE (Part 2)**
> >
> > > most experiment details related to the baseline methods are placed in the appendix ...why not add a new adapter in a new domain as in [2]?
> >
> > We already have this, which is called Post (Adapter). It is not CL because it trains different models for different domains/tasks. We can see it is much poorer than Post (RoBERTa), which post-trains an LM separately for each task. This indicates the continual post-training needs to train the full LM. We have cited this paper when we introduce Post (RoBERTa). Due to space limitations, some of the details are put in Appendix, which is also commonly done by others. But we do have summaries of them in the main text.
> >
> > > The intuitions for the methods are not very convincing
> >
> > It is hard to answer this. We respectfully ask you to provide the reasons why you think our methods are not convincing.
> >
> > > How is the model fine-tuned on the end tasks in CL baselines?...Does fine-tuning methods change the conclusion?
> >
> > All baselines are based on the full model fine-tuning. For those adapter-based baselines, both LM and adapters are trainable in fine-tuning. How fine-tuning methods change the conclusion is an interesting topic but out-of-scope of this paper. We will investigate this in future work.
> >
> > > Lack of comparison when the learning order of domains is permuted
> >
> > Please note that NCL actually gives the best on average in Table 2. (In Jin et al, (2021), NCL (called “Naive”) is also quite effective). Due to the computationally intensive nature of post-training, we chose the best baselines NCL to run different orders and compared it with our CPS. We can see the results (Table 4 in Appendix D) do not change much given different orders and CPS outperforms NCL. Since we didn’t see your review until 11/17, we can only run one more baseline before the end of stage 1. We added the KD (one of the most effective baselines in Jin’s work) results in Table 4. We can see CPS again outperforms KD in all orders. We will run the others and update them by posting new responses. They are very unlikely to do better.
> >
> > > I think the paper requires significant improvements
> >
> > We noticed there are some misunderstandings here and there, which is probably because of our writing. We further update the introduction for better understanding. We hope that you can change your mind as we believe that we have presented a novel idea, which you can also see from the review of Reviewer ApDZ.

---

> > > ### Comment · Reviewer_dAfE · 2022-11-19
> > > **Re: Response to Reviewer dAfE (Part 2)**
> > >
> > > > most experiment details related to the baseline methods are placed in the appendix ...why not add a new adapter in a new domain as in [2]?
> > >
> > > The baseline "Post (Adapter)" is not the one I am referring to. The Post (Adapter) in this paper seems like each domain has its domain-specific adapters, and for the downstream task for that domain, only the adapters for the domain is used and fine-tuned with the whole language model. In fact, I cannot see whether the fine-tuning of Post (Adapter) includes only the adapters of the specific domain or all the adapters for all the domains since this is not clearly specified in the main content and the Appendix. However, since the Post (Adapter) is used as the base line for NCL, I assume that each task only uses domain-specific adapters in downstream fine-tuning.
> > >
> > > On the contrary, the method I mentioned in [2], which is from [3], uses all adapters from all domains during fine-tuning.
> > >
> > > > The intuitions for the methods are not very convincing
> > >
> > > These problems are pointed out and explained in the other response.
> > >
> > > > Lack of comparison when the learning order of domains is permuted
> > >
> > > Thank you for your efforts. I am looking forward to the results.
> > >
> > > [3] Ruize Wang, Duyu Tang, Nan Duan, Zhongyu Wei, Xuanjing Huang, Jianshu Ji, Guihong Cao, Daxin Jiang, and Ming Zhou. 2021. K-Adapter: Infusing Knowledge into Pre-Trained Models with Adapters. In Findings of the Association for Computational Linguistics: ACL-IJCNLP 2021, pages 1405–1418, Online. Association for Computational Linguistics.

---

> > > > ### Author Response · Authors · 2022-11-23
> > > > **Re:Re: "Response to Reviewer dAfE's reponse (Part 2)"**
> > > >
> > > > > dAfE: most experiment details related to the baseline methods are placed in the appendix ...why not add a new adapter in a new domain as in [2]?
> > > > > > dAfE:The baseline "Post (Adapter)" is not the one I am referring to. The Post (Adapter) in this paper seems like each domain has its domain-specific adapters, and for the downstream task for that domain, only the adapters for the domain is used and fine-tuned with the whole language model. In fact, I cannot see whether the fine-tuning of Post (Adapter) includes only the adapters of the specific domain or all the adapters for all the domains since this is not clearly specified in the main content and the Appendix. However, since the Post (Adapter) is used as the base line for NCL, I assume that each task only uses domain-specific adapters in downstream fine-tuning. On the contrary, the method I mentioned in [2], which is from [3], uses all adapters from all domains during fine-tuning.
> > > >
> > > >
> > > > Your understanding is correct. We want to clarify this is mentioned in the text. In the first sentence of “Non-CL baselines”, we wrote “each baseline builds a separate model for each task”. This applies to (1)-(4), which means that in (4), we train a separate model (adapter) for each task. Hence, naturally, only the adapter of the specific domain is included.
> > > >
> > > > Regarding K-adapter (K is for knowledge), we apologize that we thought your description “add a new adapter in a new domain” means our baseline ‘post (adapter)’. Let us first summarize K-adapter to make sure we are on the same page. K-adapter pre-trains two different adapters, one for factual knowledge (using Wikipedia data) and the other one for linguistic knowledge (using the dependency parsing result of the Book Corpus). In fine-tuning, the output of the two adapters are concatenated to train a model for final prediction. This is entirely different from our work as we are concerned with learning a sequence of domains. Although it is possible to train a set of adapters for each domain in our case in the post-training stage, again we are not concerned with fine-tuning.
> > > > > I am looking forward to the results.
> > > >
> > > > We run all CL baselines in 5 different sequence orders. The results averaged over all domains after the final post-trained (averaged over 5 random sequences) are as follows.
> > > > | Order | 1 |  | 2 |  | 3 |  | 4 |  | 5 |  |
> > > > |---|:---:|:---:|:---:|:---:|:---:|:---:|:---:|:---:|:---:|:---:|
> > > > | Model | MF1 | Acc | MF1 | Acc | MF1 | Acc | MF1 | Acc | MF1 | Acc |
> > > > | NCL | 76.36 | 80.77 | 76.06 | 80.69 | 76.49 | 80.84 | 76.28 | 80.83 | 76.17 | 80.68 |
> > > > | NCL (Adapter) | 74.05 | 79.48 | 73.89 | 79.35 | 74.55 | 79.70 | 73.89 | 79.41 | 75.24 | 79.98 |
> > > > | DEMIX | 74.70 | 79.66 | 74.89 | 79.79 | 75.88 | 80.35 | 76.65 | 80.92 | 75.41 | 80.13 |
> > > > | BCL | 75.78 | 80.46 | 74.98 | 79.74 | 75.95 | 80.39 | 75.71 | 80.13 | 75.75 | 80.39 |
> > > > | CLASSIC | 74.63 | 79.59 | 75.56 | 80.20 | 75.15 | 80.04 | 75.24 | 80.05 | 74.55 | 79.68 |
> > > > | KD | 75.17 | 80.06 | 75.49 | 80.39 | 75.80 | 80.51 | 74.67 | 79.91 | 75.32 | 80.45 |
> > > > | EWC | 74.84 | 79.68 | 75.40 | 80.14 | 73.83 | 79.37 | 75.39 | 80.09 | 75.49 | 80.17 |
> > > > | DER++ | 75.51 | 80.30 | 75.93 | 80.67 | 77.07 | 81.02 | 76.56 | 80.99 | 76.46 | 80.85 |
> > > > | HAT | 66.80 | 75.65 | 66.39 | 74.84 | 68.23 | 75.74 | 68.36 | 75.85 | 66.74 | 75.35 |
> > > > | HAT-All | 59.93 | 71.33 | 44.28 | 62.23 | 47.43 | 63.52 | 48.95 | 64.10 | 48.10 | 63.67 |
> > > > | HAT (Adapter) | 74.63 | 79.78 | 75.10 | 79.97 | 74.33 | 79.46 | 75.09 | 79.93 | 74.88 | 79.88 |
> > > > | CPS | **77.93** | **81.91** | **76.90** | **81.10** | **76.86** | **81.09** | **77.52** | **81.65** | **78.18** | **82.10** |
> > > >
> > > > The observations are similar to those in Table 2 in the paper. We can see CPS outperforms all baselines in all task orders.

---

> > > > > ### Comment · Reviewer_dAfE · 2022-11-23
> > > > > **Re:Re:Re: "Response to Reviewer dAfE's reponse (Part 2)"**
> > > > >
> > > > > > On K-adapters
> > > > > I still think that K-adapters should be considered as a baseline. This is because while the aim of this paper is for post-training, the ultimate goal of post-training is to make the post-trained model perform well on all domains it is trained on during fine-tuning. To achieve this goal, K-adapters resort to fusing the knowledge during fine-tuning, and CPS resorts to fusing the knowledge during post-training. While the methods are different, the goal is the same. However, I can understand why the authors think this comparison is out of the scope of this paper, so I can agree not to compare with K-adapters.
> > > > >
> > > > > > On different methods for different domain orders.
> > > > >
> > > > > Thank you for the experiment. This completely addresses my concern.

---

> > ### Comment · Reviewer_dAfE · 2022-11-19
> > **Re: Response to Reviewer dAfE (Part 1)**
> >
> > > why assume "general domain datasets are not accessible to users of the LM"
> >
> > I also do not assume that we are training the language models from scratch, so there is no misunderstanding here. I also agree that the exact corpora for pre-training might not be available for the users. But my point is, **the proxy of pre-training dataset that contains general domain knowledge is always very easy to obtain**. For example, the Wikipedia dataset of different languages is a good proxy. As for the size of this general domain knowledge corpus, we can always subsample a proportion of them to an appropriate size. So using a proxy corpus may be more convincing that the initialization stage really uses the general domain knowledge.
> >
> > > I cannot agree with why using the domain-specific dataset and the loss function in Equation (4)
> >
> > The reason I cannot agree is that the robustness learned from Equation (4) is robustness learned from a domain-specific dataset. In this case, why can we say that this can make the model find the units that are important to the general knowledge? Because one can also say the robustness may be domain-specific. However, I think this question is ill-defined since the paper does not precisely define what general knowledge and domain-specific knowledge refer to. So it is hard to say what the knowledge is related to the units selected by Equation (4) and the importance score. So the easiest way I can think of to mitigate the above question is just using the proxy general domain corpora and using the MLM loss, or the loss in Equation (4), to find the important neurons.
> >
> > >why contrastive learning is used here
> >
> > The problem I am having here is: conceptually, doesn't full knowledge include learned knowledge? Then contrasting full knowledge with learned knowledge means that the full knowledge, which should contain past knowledge, should be contrasted with the past knowledge. This seems contradicting, and this is one of the parts that I find the method not very intuitive. I know that there is an ablation study that uses $1-I_{l}^{t-1}$, and it performs worse, but that actually makes me more confused about why contrasting the past and full knowledge can be useful (while it is empirically useful).
> >
> > > Why is AdapterFusion not compared here
> >
> > I don't think there is a difference between "fusing the knowledge in downstream tasks" and CF or KF. This is because we can regard different domains as different tasks, and fusing the knowledge in different domains in different tasks should mitigate CF and somewhat achieves KT.
> >
> > > this paper misses a comparison between most knowledge distillation-based methods in Jin et al. (2021), there is only one KD method in this paper.
> >
> > I apologize for not counting DER++ as a knowledge distillation method.
> >
> > > To be specific, I would like to see an ablation study that constructs a random importance score
> >
> > I appreciate the authors' effort in adding this ablation. But I find there are no details on this baseline. Can you describe more on how the random masks are selected? I want to check if this ablation is the one I am thinking of.
> >
> > > What does "plug it into the whole language model" means
> >
> > Thank you for the explanation. Then I think here, the paper can simply say "we use the LM with soft-masking" instead of using those notations and equations since they are confusing.

---

> > > ### Author Response · Authors · 2022-11-23
> > > **Re:Re: "Response to Reviewer dAfE's reponse (Part 1)" 1/2**
> > >
> > > Sorry for taking so long to respond because of the computation-intensive nature of post-training and a large number of baselines. We thank you for your patience.
> > > > the proxy of pre-training dataset that contains general domain knowledge is easy to obtain
> > >
> > > Thanks for understanding our argument and clarifying your question. To show the effectiveness of the proposed proxy KL-divergence, we agree it would be interesting to see what happens if we use a subset of Wikipedia data as the proxy dataset. We found the Wikipedia dataset in [a], which has been used to pre-train an LM and has a similar size as our domain data (around 700M). We conducted two experiments using the data: (1) CPS (Wiki+MLM) - which uses MLM as the $L_{impt}$ loss in the initialization stage to compute the importance of units (to identify the general knowledge) just like any other domains in the continual learning part, and (2) CPS (Wiki+KL), which uses KL-divergence as $L_{impt}$ in the initialization stage just like the proposed proxy method. The results are as follows (a negative forgetting rate means positive knowledge transfer):
> > >
> > > | Domain | Restaurant | Restaurant | ACL | ACL | AI | AI | Phone | Phone | PubMed | Camera | Camera | Average | Average | Forgetting Rate | Forgetting Rate |
> > > |:---:|:---:|:---:|:---:|:---:|:---:|:---:|:---:|:---:|:---:|:---:|:---:|:---:|:---:|:---:|:---:|
> > > | Model | MF1 | Acc | MF1 | Acc | MF1 | Acc | MF1 | Acc | MF1 | MF1 | Acc | MF1 | Acc | MF1 | Acc |
> > > | CPS (Wiki+MLM) | 80.22 | 87.12 | 68.12 | 72.92 | 68.55 | 76.06 | 83.50 | 86.11 | 71.94 | 86.02 | 91.15 | 76.39 | 80.88 | 0.54 | 0.40 |
> > > | CPS (Wiki+KL) | **81.25** | **87.89** | **70.89** | **74.87** | 69.68 | 76.86 | 85.98 | **87.78** | 72.03 | 86.69 | 91.44 | 77.75 | 81.81 | -0.50 | -0.27 |
> > > | CPS | 80.34 | 87.16 | 69.36 | 74.01 | **70.93** | **77.46** | **85.99** | 87.70 | **72.80** | **88.16** | **92.30** | **77.93** | **81.91** | **-1.09** | **-0.60** |
> > >
> > > We can see that CPS (Wiki + KL) is similar to CPS and outperforms CPS (Wiki + MLM). This indicates that the proposed proxy KL-divergence is more effective and data efficient. MLM actually adapts the LM to the Wikipedia data, which may not be sufficiently representative of the original data used in pre-training the LM. As a result, it ends up identifying the knowledge that is suitable only for the Wikipedia data. In contrast, the proposed proxy KL-divergence leverages the random dropout mask and measures the robustness, which is less related to a specific domain and thus reflects the (general) knowledge in the original LM better.
> > >
> > > [a]: Merity et al., Pointer Sentinel Mixture Models, ICLR 2017
> > > > The reason I cannot agree is that the robustness learned from Equation (4) is robustness learned from a domain-specific dataset.. I think this question is ill-defined since the paper does not precisely define what general knowledge and domain-specific knowledge refer to...So the easiest way I can think of to mitigate the above question is just using the proxy general domain corpora and using the MLM loss, or the loss in Equation (4), to find the important neurons.
> > >
> > > This question is highly related to the last question. The general knowledge ($G$) is just the knowledge in the pre-trained LM, which we want to preserve. The domain-specific knowledge $S_i$ of a domain $i$ is the knowledge learned by adapting the LM using the domain corpus. We measure the importance of units to $G $ by the proxy KL-divergence on robustness and measure the importance for $S_i$ by domain data and MLM loss. The final knowledge in the network is the union of $G$ and $S_i$. We protect the $G$ and $S_i$ by soft-masking the units important to them.
> > >
> > > We agree that robustness is somewhat related to the data used to compute it, but it is much less domain-specific compared to MLM because MLM directly adapts the LM to the specific domain, while the proposed method obtains two representations from random masks and minimizes the distance of two random masked representations. As we can see in the above table, CPS and CPS (Wiki+KL) are both better than CPS (Wiki+MLM).
> > > > why contrastive learning is used here
> > >
> > > Sorry for the confusion as it is quite subtle. Yes, the full knowledge includes the learned/past knowledge. Contrasting the full knowledge (including the past knowledge and the current domain knowledge) and the past (previously learned) knowledge is effective because the contrasting cannot do anything to the shared past knowledge as it is protected by soft-masks. Thus effectively, it pushes the current domain knowledge away to be complementary to the past knowledge.

---

> > > > ### Author Response · Authors · 2022-11-23
> > > > **Re:Re: "Response to Reviewer dAfE's reponse (Part 1)" 2/2**
> > > >
> > > > > dAfE: Why is AdapterFusion not compared here. One can train domain-specific adapters using unlabeled data (MLM), and use the domain-specific downstream task dataset to learn the parameter of AdapterFusion and the downstream task's parameter.
> > > > >>Authors: This paper works on continual post-training. Our focus is on how to achieve CF prevention and KT in the post-training stage. How to better fuse the knowledge in downstreaming tasks (e.g., adapterfusion) is out-of-the-scope of this paper. We have discussed and cited this paper in footnote 11.
> > > > >>> dAfE: I don't think there is a difference between "fusing the knowledge in downstream tasks" and CF or KF. This is because we can regard different domains as different tasks, and fusing the knowledge in different domains in different tasks should mitigate “CF and somewhat achieves KT.
> > > >
> > > >
> > > > Let us first summarize the AdapterFusion so that we are on the same page: AdapterFusion proposes a two-stage method to learn a set of tasks (classification tasks). In the first stage, it learns one adapter for each task independently using the task’s training data. In the second stage, it uses the training data of each task again to learn a good composition of the learned adapters in the first stage to produce the final model for each task.
> > > >
> > > > First of all, AdapterFusion is for end-task fine-tuning only. It is not directly applicable to continual post-training. If we understand you correctly, you are creating a new baseline by adapting AdapterFusion to continual post-training (*“One can train domain-specific adapters using unlabeled data (MLM), and use the domain-specific downstream task dataset to learn the parameter of AdapterFusion and the downstream task's parameter.”*). We believe you meant to fuse the post-trained adapters in **the fine-tuning stage using end-task classification data**. We said in our previous response that this is out-of-scope of this paper because we focus on **the post-training stage**. Our end-task fine-tuning is mainly for evaluation of the continually post-trained model and just uses the conventional fine-tuning method. Our paper is not concerned with fusion in fine-tuning, for which an existing method like that in AdapterFusion could be applicable. Furthermore, using a separate adapter to learn each task is also not suitable for our continual learning (CL) setting because CL usually wants different domains/tasks to share parameters rather than to build a separate model (adapter) for each task.
> > > > > I appreciate the authors' effort in adding this ablation. But I find there are no details on this baseline. Can you describe more on how the random masks are selected? I want to check if this ablation is the one I am thinking of.
> > > >
> > > > In the paper, we gave “(2) CPS (random) with randomly generated importance
> > > > scores to do soft-masking and contrastive learning.” To be more specific, we simply change Eq. 3 to $I_{l,i} = rand()$, where the $rand()$ is a function that gives random numbers from a uniform distribution on the interval $[0,1)$. All the other parts have no change (Eq. 4 can be removed).

---

> > > > > ### Comment · Reviewer_dAfE · 2022-11-23
> > > > > **Re:Re:Re: "Response to Reviewer dAfE's reponse (Part 1)" 2/2**
> > > > >
> > > > > >dAfE: Why is AdapterFusion not compared here. One can train domain-specific adapters using unlabeled data (MLM), and use the domain-specific downstream task dataset to learn the parameter of AdapterFusion and the downstream task's parameter.
> > > > > >>Authors: This paper works on continual post-training. Our focus is on how to achieve CF prevention and KT in the post-training stage. How to better fuse the knowledge in downstreaming tasks (e.g., adapterfusion) is out-of-the-scope of this paper. We have discussed and cited this paper in footnote 11.
> > > > > >>>dAfE: I don't think there is a difference between "fusing the knowledge in downstream tasks" and CF or KF. This is because we can regard different domains as different tasks, and fusing the knowledge in different domains in different tasks should mitigate “CF and somewhat achieves KT.
> > > > >
> > > > > Your understanding of my understanding is correct. I do mean to fuse the knowledge of different adapters in the end tasks. However, I see why the authors think this is out of the scope of this paper, so I agree to not compare with this baseline.
> > > > >
> > > > >
> > > > > > On the random soft-masks
> > > > >
> > > > > Thank you for the description. That is exactly the ablation experiment I am expecting. Just out of curiosity, what is the average importance score for different domains of different units of the language model?

---

> > > > > > ### Author Response · Authors · 2022-11-25
> > > > > > **Thank you very much for appreciating our work**
> > > > > >
> > > > > > Many thanks for the interesting discussion and for raising your score.
> > > > > > > Just out of curiosity, what is the average importance score for different domains of different units of the language model?
> > > > > >
> > > > > > Regarding the average importance of each domain, it varies as it is affected by the ordering of tasks due to the fact that when each task is learned, the learning shares parameters with previous tasks extensively. We computed for all 5 sequences and found that the average importance score for each domain for attention head units is 0.5433, for intermediate units is 0.4823, and for output units is 0.5256. The average scores for different sequences are similar.

---

> > > > ### Comment · Reviewer_dAfE · 2022-11-23
> > > > **Re:Re:Re: "Response to Reviewer dAfE's reponse (Part 1)" 1/2**
> > > >
> > > > Thank you for the clarifications. The supplementary experiments on the proxy dataset and MLM loss have clarified my questions on why the KL loss term can preserve the general knowledge and justify the usage of the KL loss term. I am also convinced by the explanation of why contrastive learning can be used here.

---

### Decision · Program_Chairs · 2023-01-20

**Decision:**

Accept: poster

**Justification For Why Not Higher Score:**

The original version of the paper confused the reviewers. Hopefully, the authors will polish the paper according to the discussion with reviewers.

**Justification For Why Not Lower Score:**

Overall this is a nice paper introducing a novel approach to a problem that has significant practical importance for real-world applications.

**Metareview: Summary, Strengths And Weaknesses:**

This paper studies the problem of continual post-training. Although there is previous work with the same setting, this work proposes a new solution.

This paper proposes Continual Post-training of LMs with Soft-masking (CPS) to tune pre-trained language models across different domains in a continuous learning manner. CPS is designed such that the language model can maximize the retention of knowledge learned in the general domain or previous domains. This is done by computing the importance of submodules in the transformer model and using the importance scores to mask the gradients so that critical units from previous domains do not change too much. Using the importance scores to consolidate learned knowledge is common in continual learning. But the paper further augments the typical LM Mask loss with a contrastive loss, which aims to encourage LMs to acquire knowledge from new domains. Experimental results in six domains show that the proposed method CPS outperforms other continual and non-continuous learning baselines. The reviewers suggest more baseline experiments. The additional results provided by the authors make the results more solid and convincing.

The main issue with the paper is the writing. The reviewers have questions about the intuition and technical methods in the beginning. But problems are solved after the explanation of the authors' feedback. Please carefully polish the camera-ready version based on the discussion with the authors if the paper is accepted.

**Note From Pc:**

if the above contains the word "oral" or "spotlight" please see: "oral" presentation means -> notable-top-5% and "spotlight" means -> notable-top-25%. As stated in our emails, we are disassociating presentation type from AC recommendations